# NGTTA: Non-parametric Geometry-driven Test Time Adaption for 3D Point Cloud Segmentation

## Abstract

Previous Test Time Adaption (TTA) methods usually suffer from training collapse when they are transferred to complex 3D scenes for point cloud segmentation due to the significant domain gap between the source and target data. To solve this issue, we propose NGTTA, a stable test time adaption method guided by non-parametric geometric features. In NGTTA, we leverage the distribution of non-parametric geometric features on target data as an "intermediate domain" to reduce the domain gap and guide the stable learning of the source model on target data. Specifically, we use the source domain model and a non-parametric geometric model to extract the embedding features and geometric features of the point cloud, respectively. Then, a category-balance sampler is designed to filter easy samples and hard samples in the input data to address the class imbalance issue in semantic segmentation. Inspired by previous work, we use easy samples for entropy minimization loss and pseudo-label prediction to fine-tune the source domain model. The difference is that we refine the pseudo labels not only by considering the soft voting among their nearest neighbors in the model embedding feature space but also in the geometric space, which can prevent the accumulation of errors caused by model feature shifts. Furthermore, we believe that hard samples can effectively represent the distribution differences between the source domain and the target domain. Therefore, we propose to distill the geometric features of hard samples into the source domain model in the early stages of training to quickly converge to an "intermediate domain" that is similar to the target domain. By taking advantage of the ability of the non-parametric geometric feature to represent the underlying manifolds of the target data, our method efficiently reduces the difficulty of the domain adaption. We conduct the main experiments on the more challenge ***sim-to-real*** benchmark about synthetic dataset 3DFRONT and the real-world datasets ScanNet and S3DIS for 3D segmentation task. Results show that our method can efficiently improve the mIOU by over **3%** on 3DFRONT$\rightarrow$ ScanNet and **7%** on 3DFRONT$\rightarrow$ S3DIS.

## 1 Introduction

With the development of deep learning, more and more neural networks are deployed in real-world applications. However, current deep networks may only perform optimally when the training and testing data share the same distribution (He et al., 2016; Krizhevsky et al., 2017). Therefore, deep networks often struggle to generalize on the unseen data which is known as the domain shift (Geirhos et al., 2018; Hendrycks & Dietterich, 2019; Recht et al., 2019). Unsupervised Domain Adaptation (UDA) techniques have emerged as a popular solution for addressing domain shift in deep learning (Saito et al., 2017; Peng et al., 2021; Wu et al., 2019; Long et al., 2017). These methods aim to transfer knowledge from a labeled source domain to an unlabeled target domain during training. While UDA methods have demonstrated effectiveness in improving the performance of deep networks in the presence of domain shift, a key limitation is the requirement to have knowledge of the test data during training. This constraint can greatly diminish the practical utility of UDA techniques.

Recently, an interesting and practical paradigm known as Test Time Adaption (TTA) is attracting more and more attention (Wang et al., 2021; Chen et al., 2022; Niu et al., 2023; Zhang et al., 2022a).

Figure 1: llustration of the overall pipeline. **i)** In Stage 1, the model was trained on the source domain with the given label. From the source distribution, we can see that the class distribution is better separated from each other due to the correct supervision signal. **ii)** In Stage 2, due to the significant differences in feature distribution, transferring directly from S2T-Distribution to Target-Distribution based on previous TTA methods is difficult. (For example, class distribution inside the yellow box in Target Distribution overlaps almost completely in S2T-Distribution). **iii)** We use the non-parametric geometric feature, which can be considered as the 'intermediate domain' between the source domain and the target domain to guide the adaptation process.

It does not need to access the training method and training data used by the model and can adapt any trained source model to the test data in testing time. This flexibility and independence make TTA a valuable approach for addressing domain shift in real-world scenarios. However, previous TTA methods often perform training collapse on the 3D segmentation task, especially on the more challenging sim-to-real benchmark proposed in (Ding et al., 2022). By digging into the failure cases, we found the two main challenges: i) Previous TTA methods often use entropy minimization loss on classification tasks, but when this loss is applied to segmentation tasks, it may cause the model to be overconfident in the majority class due to the more serious class imbalance problem. ii) 3D scenes exhibit higher complexity compared to images, and the distribution variations among different scenes are significantly greater, so using a simple TTA method to complete this difficult process may lead to the training collapse, which is shown in Figure 1.

The first challenge arises mainly because previous TTA methods set a fixed low threshold to select low-entropy easy samples for entropy minimization loss. However, due to the common issue of class imbalance in semantic segmentation, low-entropy samples are predominantly found in the major categories, which exacerbates the class imbalance problem further. To address this issue, we propose a category-balance sampler. Unlike previous methods that set a fixed threshold, we calculate a corresponding entropy threshold for each category based on its sample count. This approach reduces the differences in the number of easy samples across categories, effectively mitigating the class imbalance problem. Then, we propose to use easy samples for entropy minimization loss and pseudo-label prediction loss.

To solve the second challenge, we hope to find an "intermediate domain" between the source domain and target domain to guide the adapting process of the model thereby promoting training stability. Inspired by (Ran et al., 2022; Sun et al., 2024), explicit geometric representations can capture the underlying manifolds of the data, thereby offering insights into the rough distribution of the target domain to improve the model's generalization to unseen data. Therefore we use the non-parametric geometric feature as the "intermediate domain" between the source domain to the target domain which is shown in Figure 1. Specifically, we additionally utilize a non-parametric geometric model to obtain the geometric features of the point cloud. First, for the easy samples, we recognize that the source domain model has a feature-shifting problem on the unseen data, which may lead to incorrect pseudo-labels and subsequently cause the accumulation of errors. Therefore, we refine the pseudo labels not only by considering the soft voting among their nearest neighbors in the model's embedding feature space but also in the geometric space. Then, we believe that hard samples can effectively represent the differences between the source domain and the target domain. Therefore, we distill the geometric features of the hard samples into the source domain model. This process helps the source domain model quickly learn the underlying manifold distribution of the target domain, converging to an "intermediate domain" that closely resembles the target domain. This approach enhances the stability and performance of domain adaptation.

Testing time adaptation for indoor 3D scene segmentation tasks remains an unexplored area. To the best of our knowledge, we are the first to attempt such an approach. Therefore, we propose to follow

the UDA (Unsupervised Domain Adaptation) method Ding et al. (2022) and conduct experiments using the challenging sim2real benchmark. Our method can efficiently improve the source model by over 3% mIOU on 3DFRONT→ScanNet and 7% on 3DFRONT→S3DIS. Our contributions can be summarized as follows:

- We proposed a category-balance sampler to filter easy samples and hard samples, ensuring that the difference in the number of positive samples for each category is minimized, thereby addressing the class imbalance problem.

- We propose to leverage the non-parametric geometric feature as the "intermediate domain" to stabilize the adaption process to rapidly converge the model to a distribution that approximates the target domain.

- We conducted experiments on the challenging sim2real benchmark, and the competitive experimental results validate the effectiveness of our method.

## 2 RELATED WORK

**Unsupervised Domain Adaptation:** UDA (Saito et al., 2017; Peng et al., 2021; Wu et al., 2019; Long et al., 2017; Yang et al., 2020; Zou et al., 2018; Cui et al., 2020) aims at transferring knowledge of the labeled source domain to the unlabeled target domain in the training time. Segmentation tasks are more difficult to perform domain generalization than simple classification tasks, especially on the 3D data. Zou et al. (2018) improved the performance by solving the class imbalance problem in the segmentation task. Squeezesegv2 (Wu et al., 2019) consider the density and geometric during the domain adaption. Jaritz et al. (2020) propose to leverage the information of images and point cloud to complete multi-modality UDA.

**Test Time Adaption:** Almost all previous TTA Methods have been applied to classification tasks. Wang et al. (2021) firstly proposes fullly test time adaption which does not need to access the training method and training data and uses entroy minimization loss to optimize the model. Then, many subsequent works (Zhao et al., 2023; Niu et al., 2023; Zhang et al., 2022a; Wang et al., 2022; Niu et al., 2022) modify the entropy loss to further improve the performance. Chen et al. (2022) leverages self-supervised contrastive learning and a soft voting strategy for refining the pseudo-label to facilitate target feature learning. Iwasawa & Matsuo (2021) tries to update the category prototypes on the target domain to provide a more accurate decision boundary. Howevere, there have been few test time adaption methods focusing on the segmentation task, especially on the more difficult 3D data. Song et al. (2023) firstly explore TTA method for segmentation in the dynamic world. Shin et al. (2022) propose a multi-modal test time adaption framework for 3D segmentation. However, the need for multi-modal limits its general applications. Therefore, it is necessary to develop a general and effective TTA method for 3D segmentation.

**3D Point Cloud Segmentation:** Due to the disorder and irregularity of the point cloud data, Qi et al. (2017) firstly propose PointNet++ to use ball query or kNN to construct local neighborhood and aggregate it by symmetry pooling. Many subsequent works (Qian et al., 2022; Lin et al., 2023; Thomas et al., 2019; Zhao et al., 2021) design more complex modules to extract the local feature based on the PointNet++. PointTransformer (Zhao et al., 2021) leverage the local attention mechanism to extract local feature. KPConv (Thomas et al., 2019) defined the anchor points and used them to compute the aggregate weight. In addition, Ran et al. (2022); Sun et al. (2024) propose to leverage the explicit geometric to introduce strong prior which can reduce the learning difficulty and improve the performance. Furthermore, PointNN (Zhang et al., 2023) uses trigonometric functions to capture the non-parametric geometric feature for point cloud recognition which has proved the strong generalization ability to unseen 3D data. Inspired by the above methods, we decided to leverage the non-parametric geometric feature to prompt the learning of test time adaption.

## 3 PROPOSED METHOD

We address the closed-set fully test time adaption in 3D segmentation task that we can only access the source model during the adaption process. As shown in Figure 1, the source model is trained on the labeled source domain $\{x_i^s, y_i^s\}_{i=1}^{N_s}$, where $x_i^s$ is the input $i$-th scene in source dataset, $y_i^s$ is the corresponding label, and $N_s$ is the total number of scenes in source dataset. The goal of

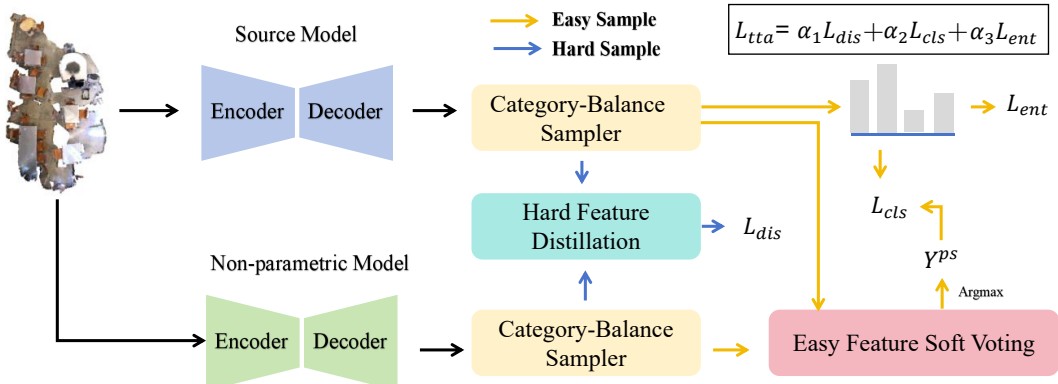

Figure 2: llustration of the proposed framework. We use a Category-Balance Sampler to balance the selection of simple and hard samples for each category. Simple samples are used for entropy minimization loss and pseudo-label prediction loss, with pseudo-labels optimized through soft voting using features from the non-parameterized geometric model and the source domain model. hard samples are used to distill their geometric features into the source domain model.

test time adaption is to adapt the source model on the target domain $\{x_i^t\}_{i=1}^{N_t}$ without accessing its labels $\{y_i^t\}_{i=1}^{N_t}$ during the adaption process, where $N_t$ is the total number of scenes in the target dataset. It's worth noting that our setting is closed-set which means that both source domain and target domain share the same semantic classes. Therefore, there will be $y_i^s = 0, 1, ..., N_c - 1$ and $y_i^s = 0, 1, ..., N_c - 1$, where $N_c$ is the total number of classes. Since test time adaption only considers training on the target domain, we use $x_i$ and $y_i$ to replace the $x_i^t$ and $y_i^t$ for convenience and the following symbols are all defined on the target domain.

The framework of NGTTA is shown in Figure 2. Firstly, We designed a category-balance sampler to balance the selection of easy samples and difficult samples and use easy samples to perform entropy minimization loss and pseudo-label prediction loss, which will be introduced in 3.1. Then, in 3.2 we additionally propose to use a non-parametric geometric model to extract the geometric information of point clouds. For easy samples, we refine the pseudo-labels by modified soft voting, which averages the neighborhood predictions from both the model feature space and the geometric feature space. For difficult samples, we distill their geometric features into the source domain model, helping the source domain model quickly converge to an "intermediate domain" that is similar to the target domain.

## 3.1 CATEGORY-BALANCE SAMPLER

Our method is generally applicable to point-based models. Therefore, for any model $\mathcal{M}^s$ that is trained on the source domain, the output for the input scene $x_i \in R^{M \times 3}$ in the target domain will be the point-wide embedding feature $f_i \in R^{M \times C}$, where $M$ represents the number of points in $i$-th scene and $C$ represents the number of feature channel.

$$f_i = \mathcal{M}^s(x_i) \tag{1}$$

Then, $f_i$ is sent to the classification head to produce the class probability $p_i \in R^{M \times N_c}$.

$$p_i = Head(f_i) \tag{2}$$

We calculate the entropy $E_i$ of the samples based on the class probability $p_i$, which can represent the confidence level of the model's predictions.

$$E_i = -\sum_{c=0}^{N_c-1} p_i[c] \log p_i[c] \tag{3}$$

Next, we need to determine whether each sample is an easy sample or a hard sample based on its entropy and the corresponding threshold. Previous methods shared the same threshold for samples

across all categories to select easy low-entropy samples. However, indoor scene segmentation generally suffers from class imbalance issues, where the majority of confidently predicted low-entropy samples are found in the major categories. As a result, samples from tail categories are often difficult to select, which further exacerbates the class imbalance problem.

Therefore, we propose a category-balance sampler. Specifically, we first calculate the number of samples $Z_c$ for the $c$-th category and define the category with the highest sample count as the major category, with the count being $Z_m$. Then, we define the threshold for $c$-th category as follows:

$$\sigma_c = \sigma + (1 - \frac{Z_c}{Z_m})\gamma \tag{4}$$

where $\sigma$ is the initial threshold, and $\gamma$ adaptively adjusts the threshold based on the number of samples in each category. We can see that as the sample count decreases, the threshold will gradually increase. This means that tail categories with fewer samples will have a larger entropy threshold, allowing for the selection of more samples to reduce the disparity between the number of samples across different categories.

Then, we define the set of easy samples $\mathcal{G}_i^e$ as follows:

$$\mathcal{G}_i^e = \{j \mid E_{ij} < \sigma_{y'_{ij}}\} \tag{5}$$

where $j$ means the $j$-th sample in the $i$-th scene. $y'_{ij}$ represents the class prediction of the $j$-th sample, where $y'_{ij} = Argmax(p_{ij})$.

In contrast, the set of hard samples $\mathcal{G}_i^h$ is defined as follows:

$$\mathcal{G}_i^h = \{j \mid E_{ij} \geq \sigma_{y'_{ij}}\} \tag{6}$$

Inspired by previous TTA methods Wang et al. (2021), we apply entropy minimization loss to low-entropy samples to avoid increasing the confidence level of incorrect predictions, which can be written as follows:

$$L_{ent} = \underset{j \in \mathcal{G}_i^e}{Min}(E_{ij}) \tag{7}$$

However, entropy minimization loss is category-agnostic and can only enhance the model's confidence level. To improve the model's performance on semantic segmentation metrics, we introduce pseudo-label prediction loss. Simply put, we copy the source domain model as a momentum model to predict pseudo-labels. Unlike the source domain model, we fix the parameters of the momentum model during training, and every $n$ epochs, we copy the parameters from the source domain model to the momentum model to ensure the stability of the pseudo-labels. We then use the class predictions from the source domain model together with the pseudo-labels to compute the classification loss. The process can be written as follows:

$$L_{cls} = \underset{j \in \mathcal{G}_i^e}{CrossEntropy}(p_{ij}, y_{ij}^m) \tag{8}$$

where $y_{ij}^m$ means the pseudo-label from the momentum model.

## 3.2 NON-PARAMETRIC GEOMETRY-DRIVEN ADAPTION

As discussed above, due to the significant domain gap between source and target in 3D segmentation task, the adaption process from source to target is difficult. Some domain adaption methods (Li et al., 2021; Wang et al., 2023) propose to define an "intermediate domain" that guides the source model to progressively adapt to the target domain. However, those methods need to access the source data, which is not suitable for our setting. Inspired by (Ran et al., 2022; Sun et al., 2024), non-parametric geometric feature can capture the underlying manifolds of point cloud data which can represent the rough approximation of the target distribution, so we leverage non-parametric geometric feature as the "intermediate domain" to boost the test time adaption for 3D segmentation.

We leverage PointNN (Zhang et al., 2023) as the non-parametric model which has a strong generalization ability from seen to unseen 3D data. For PointNN, it uses farthest point sampling and kNN to downsample and construct the neighborhood like PointNet++ (Qi et al., 2017). However, PointNet++

uses learnable MLP to extract neighborhood point features, while PointNN uses non-parametric trigonometric functions. Since it does not require training, PointNN can be directly applied to unlabeled test time adaption process.

Specifically, for the input scene $x_i$, non-parametric model $\mathcal{M}^g$ output the geometric feature $f_i^g$, which can be written as follows:

$$f_i^g = \mathcal{M}^g(x_i) \tag{9}$$

*Easy Feature Soft Voting.* Although easy samples have a higher confidence level, there are still incorrect predictions that lead to the provision of erroneous pseudo-labels. To address this issue, we propose constructing K-nearest neighbors using the features of easy samples and correcting the pseudo-labels through soft voting based on the neighbor class predictions.

However, due to the feature shift phenomenon of the source domain model in the target domain, the K-nearest neighbors may not be accurate, which affects the correction of the pseudo-labels. To address this issue, we simultaneously consider K-nearest neighbors constructed using geometric features during the soft voting process. Specifically, for the $j$-th easy sample feature of source model and the non-parametric model $f_{ij}$ and $f_{ij}^g$, where $j \in \mathcal{G}_i^e$. We construct the neighborhood $Q_{ij}$ and $Q_{ij}^g$ by kNN. Then we use the $y_i^m$ from momentum model to generate the new pseudo-label, which can be written as follows:

$$Y_{ij}^{ps} = \beta_t \frac{1}{K} \sum_{k \in Q_{ij}} y_{ik}^m + (1 - \beta_t) \frac{1}{K} \sum_{k \in Q_{ij}^g} y_{ik}^m \tag{10}$$

where $Y_{ij}^{ps}$ is the refined pseudo-label of the $j$-th sample in $i$-th scene. And the $\beta_t$ is the weight factor that gradually changes as training progresses. In simple terms, we believe that as the source domain model gradually converges to the target domain, the feature shift phenomenon decreases, and thus the importance of geometric features will diminish. Therefore, we assign higher weight to geometric features in the early stages of training, while in the later stages, we assign higher weight to the model's embedding features.

$$\beta_t = \frac{t}{T} \beta \tag{11}$$

where $t$ is the current training step and $T$ is the total number of training steps.

Finally, we modify Eq. 8 as follows to achieve better classification loss.

$$L_{cls} = \underset{j \in \mathcal{G}_i^e}{CrossEntropy}(p_{ij}, Y_{ij}^{ps}) \tag{12}$$

*Hard Feature Distillation.* Previous methods typically consider hard samples as noisy samples, thus only handling easy samples while discarding hard ones. However, we believe that many hard samples arise from the significant differences between the target domain and the source domain in the context of scene segmentation sim2real benchmarks, and therefore they can represent the information about the distributional differences. To this end, we decide to utilize this difference information to facilitate the adaptation of the source domain model to the target domain. Rather than using the erroneous predictions of hard samples for the aforementioned two losses, we believe that distilling their features to transfer distributional information is a more effective approach.

Therefore, we propose a feature distillation loss aimed at distilling geometric features into the source domain model, enabling it to quickly converge to an "intermediate domain" that is closer to the target domain, thereby stabilizing the adaptation process, which can be written as:

$$L_{dis} = \sum_{j \in \mathcal{G}_i^h} MSE(MLP(f_{ij}), f_{ij}^g) \tag{13}$$

where $MLP$ is a multil-ayer perceptron used to align the dimensions of source model feature with the geometric feature.

## 4 EXPERIMENTS

### 4.1 DATASETS

We conducted experiments on three datasets consists of a synthetic dataset 3DFRONT and the real-world datasets ScanNet and S3DIS for 3D segmentation task.

Table 1: Test Time Adaption Results of Sim-to-Real (3DFRONT→ScanNet and 3DFRONT→S3DIS) Benchmark . We report the mIOU (%). mACC (%) and OA(%) of different UDA and TTA methods. **Bold** represents the best performance in UDA and TTA methods

| Type | Method | 3DFRONT→ScanNet | | | 3DFRONT→S3DIS | | |
|---|---|---|---|---|---|---|---|
| | | mIOU | mACC | OA | mIOU | mACC | OA |
| | Source Only | 34.80 | 49.24 | 71.66 | 29.38 | 39.72 | 68.47 |
| UDA | SqueezeSegV2 (Wu et al., 2019) | 34.98 | 49.72 | 71.89 | 29.80 | 40.12 | 69.81 |
| | AdaptSegNet (Tsai et al., 2018) | **40.23** | **52.16** | **75.10** | **37.12** | **49.82** | **76.23** |
| | APO-DA (Yang et al., 2020) | 37.82 | 50.92 | 73.27 | 35.21 | 47.69 | 74.91 |
| TTA | TENT (Wang et al., 2021) | 15.63 | 28.62 | 55.41 | 31.98 | 42.38 | 71.23 |
| | DOT (Zhao et al., 2023) | 18.30 | 29.71 | 56.58 | 32.61 | 43.41 | 72.30 |
| | AdaContrast (Chen et al., 2022) | 30.57 | 49.61 | 73.05 | 33.28 | 45.20 | 73.01 |
| | MEMO (Zhang et al., 2022a) | 14.21 | 27.93 | 54.96 | 27.10 | 37.98 | 67.10 |
| | T3A (Iwasawa & Matsuo, 2021) | 17.20 | 29.17 | 56.02 | 32.11 | 43.11 | 71.80 |
| TTA | **Ours** | **38.42** | **51.30** | **74.03** | **36.36** | **49.07** | **75.74** |

**ScanNet** is proposed in (Dai et al., 2017), which is a popular real-world 3D scene dataset with 1,201 scans for training, 3,12 scans for validation and 100 scans for testing. It has rich dense segmentation annotations for 20 categories.

**S3DIS** is proposed in (Armeni et al., 2016), which is a real-world 3D scene dataset with 271 scenes and rich dense segmentation annotations for 13 categories. Following the previous work (Qi et al., 2017), we used Area5 as validation set and others as training set.

**3DFRONT** is proposed in (Fu et al., 2021a), which consists of 13,151 CAD 3D objects in 18,968 rooms from the synthetic datasets (Fu et al., 2021b). We follow the setting in (Ding et al., 2022) which selects 4995 rooms as training samples after filtering out noisy rooms. Since it is the synthetic dataset, the scene in 3DFRONT is usually complete and easy, while the real world usually has the problems of missing data and noise.

**Closed-Set Setting.** Test Time Adaption is the closed-set setting that the source and the target domain share the same categories. Therefore, we follow the setting in (Ding et al., 2022) to select 11 categories for 3DFRONT→ ScanNet and 3DFRONT→ S3DIS. To better demonstrate the generality of our approach, we also select 8 categories for the domain adaption between ScanNet and S3DIS. **The selected categories are all the categories that are shared between the two datasets..**

### 4.2 IMPLEMENTATION

**BackBone.** To prove the effectiveness of our method, we used the state-of-the-art model Point-Meta (Lin et al., 2023) as the source model by default in the following experiments. We also reported the performance of other models to demonstrate the applicability of our method.

**Optimizable Parameters.** How to determine the optimal parameters is important in test time adaption. Previous research denotes that the knowledge of data domain is saved in "BatchNorm". Therefore, TENT propose to only update the BatchNorm Parameters. However, due to the complexity of 3D data, scenarios may not be independently and identically distributed among themselves, which greatly affects the performance of optimizing BatchNorm. AdaContrast (Chen et al., 2022) use contrast learning to optimize the entire model. However, in 3D segmentation task, the model is also more complex than the classification model, which is often the "Encoder-Decoder" architecture that the optimization is very difficult. Therefore, in our implementation, we only optimized the embedding layer at the beginning of the model and the classification head at the end.

### 4.3 RESULTS

**Sim-to-Real Benchmark** We tested different UDA and TTA methods on this benchmark which is shown in Table 1. We see that due to the significant domain gap between the synthetic and real-world datasets, the performance of the source model is only about 30% mIOU. We first report the current

Table 2: Test Time Adaption Results of Cross-site Benchmark . We report the mIOU (%) of different UDA and TTA methods.

| Type | Method | S3DIS→ScanNet | | | ScanNet→S3DIS | | |
|---|---|---|---|---|---|---|---|
| | | mIOU | mACC | OA | mIOU | mACC | OA |
| | Source Only | 48.20 | 65.01 | 78.18 | 50.99 | 61.65 | 77.66 |
| UDA | SqueezeSegV2 (Wu et al., 2019) | 46.31 | 63.17 | 76.27 | 51.20 | 62.12 | 77.93 |
| | AdaptSegNet (Tsai et al., 2018) | 50.34 | 66.12 | 79.20 | 50.12 | 60.89 | 76.23 |
| | APO-DA (Yang et al., 2020) | **53.35** | **68.87** | **81.92** | **52.65** | **63.01** | **78.62** |
| TTA | TENT (Wang et al., 2021) | 51.49 | 67.01 | 80.22 | 50.21 | 61.01 | 77.23 |
| | DOT (Zhao et al., 2023) | 51.92 | 67.51 | 80.67 | 51.00 | 61.72 | 77.92 |
| | AdaContrast (Chen et al., 2022) | 52.12 | 67.98 | 81.21 | 52.35 | 62.71 | 78.01 |
| | MEMO (Zhang et al., 2022a) | 47.23 | 64.65 | 77.12 | 49.97 | 60.92 | 76.92 |
| | T3A (Iwasawa & Matsuo, 2021) | 51.01 | 66.99 | 79.83 | 50.61 | 61.21 | 77.68 |
| TTA | **Ours** | **52.71** | **68.23** | **81.62** | **53.78** | **64.25** | **79.34** |

UDA methods, because the target domain data is accessed during training, the UDA method can effectively improve the performance, especially AdaptSegNet (Tsai et al., 2018) improving the mIOU about 5.43% and 7.74% of 3DFRONT→ScanNet and 3DFRONT→S3DIS. On the contrary, due to the complexity of 3D scenes, as well as the sim-to-real difficulty, the TTA method does not perform optimally. For example, the miou of the most classical TENT method decreases on the 3DFRONT→ScanNet respectively about 19.17%. Afterward, through the experimental results, we find that DOT (Zhao et al., 2023) and AdaContrast (Chen et al., 2022) generally have better performance, because the former considers the class imbalance problem, and the latter introduces contrastive learning and pseudo-label adjustment strategies. Furthermore, we find that the TTA method generally performs better on the simpler 3DFRONT→S3DIS. That is because the scene of S3DIS is more complete and easy than ScanNet and the domain gap with 3DFRONT is smaller. According to the results of previous methods, we believe that reducing the domain gap and solving the class imbalance is an effective way to improve the performance of TTA. Therefore, by introducing non-parametric geometric feature as "intermediate domain" and the class-balance entropy minimization loss, our method can improve the mIOU about **3.62%** and **6.98%** of 3DFRONT→ScanNet and 3DFRONT→S3DIS, which is much better than the previous TTA method and is competitive with the UDA method.

**Cross-Site Benchmark** We also tested our method on the cross-site benchmark which consists of S3DIS→ScanNet and ScanNet→S3DIS. The results are shown in Table 2. We see that due to the smaller domain gap of cross-site than the sim-to-real benchmark, TTA methods perform optimally, which proves the above conclusion. Similarly, DOT and AdaContrast perform well on this benchmark because their focus is more suitable for segmentation tasks. Our method can improve mIOU about 4.51% and 2.77%. These experiments demonstrate the effectiveness of our method on multiple benchmarks and demonstrate the applicability of our method.

Table 3: Test Time Adaption Results of Different Backbones. We use our method on different models of the 3DFRONT→S3DIS.

| Models | mIOU | GFLOPs | TP |
|---|---|---|---|
| PointMetaBase-L (Lin et al., 2023) | 29.38 | 2.0 | 192 |
| **+ours** | 36.36 | 2.9 | 178 |
| PointNet++ (Qi et al., 2017) | 22.74 | 7.2 | 181 |
| **+ours** | 28.10 | 7.9 | 171 |
| PointTransformer (Zhao et al., 2021) | 25.32 | 2.80 | 170 |
| **+ours** | 32.11 | 3.72 | 156 |
| PointNeXt-L (Thomas et al., 2019) | 27.15 | 15.2 | 126 |
| **+ours** | 33.26 | 16.1 | 118 |

Table 4: Comparison with pre-trained models.**NP** means non-parametric model and **PT** means pretrained model.

| type | Pretrain Dataset | Models | mIOU |
|---|---|---|---|
| NP | - | PointNN | 36.36 |
| PT | ScanNet | PointM2AE | 34.92 |
| | | CSC | 35.31 |
| | | MSC | 35.52 |
| | Structure3D | PointM2AE | 36.13 |
| | | CSC | 36.02 |
| | | MSC | 36.28 |

**More BackBones and Efficiency.** To prove the generality of our method, we conducted the experiments of different backbones on the 3DFRONT→S3DIS which is shown in Table 3. We tested four backbones which consist of PointMeta, PointNet++, PointTransformer and PointNeXt. The

results show that our method is applicable to most point-based models and can effectively enhance their performance, demonstrating the versatility of our approach. In addition, we also report the efficiency of NGTTA, including the computational cost in GFLOPs and the inference speed measured in Throughput (ins./sec). From the results, it can be observed that the additional computational overhead we introduced is acceptable compared to the original model's expenses. This is mainly attributed to two design features: **1)** We modified the original PointNN to reduce the calculation cost by reducing the number of neighborhood points, feature dimensions and layers, which will be introduced in supplementary material. **2)** In the computationally intensive soft voting component, we only selected a small proportion of clean samples to perform the operation, significantly reducing the computational overhead.

## 4.4 ABLATION STUDY

**Comparison with pre-trained models** In this section, we compare the performance of using parameterized pre-trained models (PointM2AE Zhang et al. (2022b), CSC Hou et al. (2021), MSC Wu et al. (2023)) and non-parametric models to drive domain adaptation, which can be seen in Table 4. The results indicate that the performance of the non-parametric model is superior. We believe this is primarily due to two reasons: **1)** The high-dimensional feature representation of parameterized models is more abstract, which makes it difficult to facilitate the rapid convergence of the source domain model. **2)** Pre-trained models still learn information specific to the dataset, which can affect the domain adaptation process. As the scale of the pre-training data increases, this influence gradually diminishes. For example, models pre-trained on Structure3D perform better than those pre-trained on ScanNet. Therefore, compared to pre-trained models, non-parametric models can more explicitly represent the features of the current data. Additionally, they do not contain any specific dataset information and require no training steps. Thus, at the current stage, non-parametric models still outperform parameterized pre-trained models.

**Different Parts in Our Method.** We tested the different parts in our method on the 3DFRONT→S3DIS benchmark, which is shown in Table 5. There are three main parts in our method, which consists of Distill (distillation from the non-parametric model), Soft Voting and Category-Balance Sampler. From the results we see that all three modules contribute to the performance improvement, of which Distill has the most significant improvement due to the significant reduction of the domain gap by utilizing "intermediate domains". Soft Voting and Category-Balance Sampler can effectively improve the performance by more than 1% mIOU because they provide more accurate pseudo-labels and solve the class imbalance problem.

Table 5: Ablation study result of different parts in our method. We use PointMetaBase-L as the source model and test it on the 3DFRONT→S3DIS.

| Distill | Soft Voting | Category-Balance Sampler | mIOU |
|---------|-------------|--------------------------|------|
|  |  |  | 29.38 |
| ✓ |  |  | 32.28 |
| ✓ | ✓ |  | 34.92 |
|  | ✓ |  | 31.76 |
|  |  | ✓ | 30.62 |
| ✓ | ✓ | ✓ | **36.36** |

Table 6: Ablation Study of Soft Voting on the 3DFRONT→S3DIS benchmark.

| Distance | Neighbor Numbers | mIOU |
|----------|------------------|------|
| Feat | 2 | 35.12 |
|  | 10 | 35.83 |
|  | 20 | 35.01 |
| Feat+Geo | 2 | 35.43 |
|  | 10 | 36.36 |
|  | 20 | 36.21 |

**Soft Voting.** We conducted the ablation study of soft voting in the Table 6. First, we tested soft voting for building neighborhoods with only the features of the model (**Feat**) which is proposed in AdaContrast (Chen et al., 2022). We see that when the number of neighborhoods is 10, the performance is the best, which also proves the importance of the soft voting strategy. However, when the number of neighborhoods continues to increase, the offset model features may introduce neighborhood samples that are not similar, resulting in wrong pseudo-labels. On the contrary, when non-parametric geometric feature is added to construct the neighborhood, the performance is usually higher, and due to the stable geometric features on unseen data, the performance is still good even if the number of neighborhoods is increased, which indicates the stability of our soft voting strategy.

**Entropy Threshold.** In this part, we explore the impact of $\sigma$ and $\gamma$ on performance which is shown in Table 7 and Table 8. We use PointMetaBase-L as the source model and test it on the

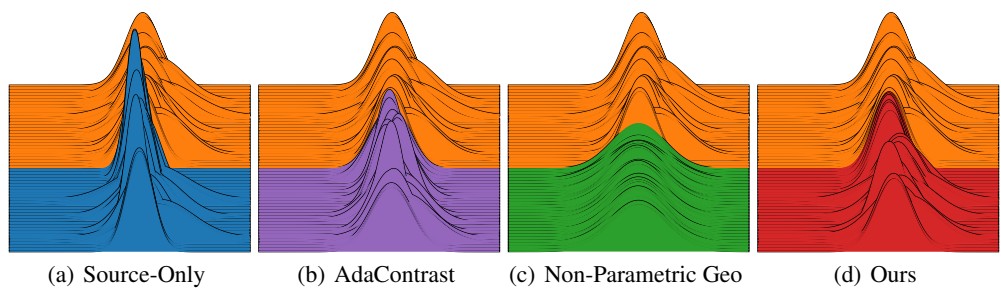

(a) Source-Only  (b) AdaContrast  (c) Non-Parametric Geo  (d) Ours

Figure 3: Visualization of Feature Distribution on the 3DFRONT→S3DIS benchmark. **Front** is the distribution of the source model with different methods. **Back** is the distribution of the model trained by labels on target domain.

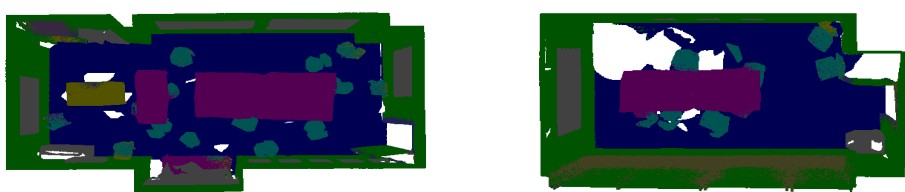

Figure 4: Visualization of Segmentation Results on the 3DFRONT→S3DIS benchmark.

3DFRONT→S3DIS. In Table 7, we fixed $\gamma$=0.3 and observed that a smaller threshold effectively filters out difficult samples that are prone to generating erroneous predictions, thereby improving the model's performance. Similarly, in Table 8, we fixed $\sigma = 0.2$. And when set $\gamma = 0$, it indicates that all categories share the same threshold. We observed a significant drop in performance, which demonstrates the severe impact of class imbalance on performance.

<table>
<tr><td colspan="6">Table 7: Ablation Study of the $\sigma$</td></tr>
<tr><td></td><td>0.1</td><td>0.2</td><td>0.3</td><td>0.5</td><td>0.6</td></tr>
<tr><td>$\sigma$</td><td>36.18</td><td>36.36</td><td>36.01</td><td>35.91</td><td>35.52</td></tr>
</table>

<table>
<tr><td colspan="6">Table 8: Ablation Study of $\gamma$</td></tr>
<tr><td></td><td>0</td><td>0.1</td><td>0.2</td><td>0.3</td><td>0.4</td></tr>
<tr><td>$\gamma$</td><td>34.92</td><td>35.63</td><td>35.85</td><td>36.36</td><td>36.12</td></tr>
</table>

## 4.5 VISUALIZATION

**Feature Distribution.** We visualized the distribution of features in Figure 3. In Figure 3(a), we visualize the distribution of source domains and target domains and see that they have a very large domain gap. In Figure 3(c), we visualize the non-parametric geometric feature distribution and the target domain distribution. We can see that they capture the information of the target domain distribution to a certain extent, such as the height between samples in their domain are relatively close. In Figure 3(d), the distribution of the source domain adjusted by our method is basically similar to that of the target domain, which effectively proves the effectiveness of our method.

**Segmentation Results.** We visualized the segmentation results in Figure 4 on the 3DFRONT→S3DIS benchmark.

## 5 CONCLUSION

We argue that non-parameter geometric features can capture the underlying manifold of unseen data, which has strong generalization. Therefore, we leverage non-parametric geometry as an intermediate domain to prompt test time adaption. By introducing distillation from non-parametric model, pseudo-label refined by soft voting and category-balance sampler, our method can effectively improve the performance of the source domain model in the target domain.

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

# A    APPENDIX

## A.1    LIGHTWEIGHT POINTNN

To reduce computational overhead, we modified the settings of PointNN by decreasing the number of layers ($\mathbf{L}$), the number of neighboring points ($\mathbf{K}$), downsample ratio ($\mathbf{R}$), and the init feature dimensions $\mathbf{C}$ to enhance the efficiency of NGTTA. Although the modifications made to PointNN may result in a slight decrease in accuracy, the resulting improvements in efficiency are significant. This enhances the applicability of NGTTA in real-world scenarios.

- PointNN: L=5,K=90,R=2,C=144

- Ours: L=4,K=24,R=4,C=36

## A.2 ABLATION STUDY OF THE LOSS WEIGHT $\alpha$

The overall loss definition of NGTTA is as follows:

$$L_{tta} = \alpha_1 L_{dis} + \alpha_2 L_{cls} + \alpha_3 L_{ent} \tag{1}$$

where $\alpha_1$, $\alpha_2$ and $\alpha_3$ mean the loss weight of $L_{dis}$, $L_{cls}$ and $L_{ent}$.

In the following three tables, we use PointMetaBase-L as the source model and test the impact of the loss weight on the 3DFRONT$\rightarrow$ S3DIS. From the results, we can see that when the weight is set to 0, there is a decline in performance, which demonstrates the necessity of each loss component. At the same time, we found that the performance does not fluctuate significantly around the optimal weight value, indicating that our method is not particularly sensitive to the loss weights, thereby proving the stability of the approach.

Table 1: Ablation Study of $\alpha_1$

| | 0 | 1 | 10 | 100 |
|---|---|---|---|---|
| $alpha_1$ | 33.81 | 34.96 | 36.36 | 36.01 |

Table 2: Ablation Study of $\alpha_2$

| | 0 | 0.1 | 0.5 | 1 |
|---|---|---|---|---|
| $\alpha_2$ | 34.10 | 35.89 | 36.36 | 36.21 |

Table 3: Ablation Study of $\alpha_3$

| | 0 | 0.1 | 0.5 | 1 |
|---|---|---|---|---|
| $\alpha_3$ | 34.60 | 35.91 | 36.30 | 36.36 |

## A.3 ABLATION STUDY OF OPTIMIZABLE PARAMETERS

In our implementation, we only optimized the initial layers of the encoder and the classification layer. In this section, we explore the impact of different optimization parameters on performance. We set up various combinations of optimizable parameters: 1) Optimize only batch normalization parameters (**BN**) 2) Optimize the entire model (**AM**) 3) Optimize only the classifier (**CH**) 4) Optimize only the encoder (**EC**) 5) Optimize only the decoder (**DC**) 6) Our implementation (**Ours**) 7) Optimize the encoder and the decoder (**BC**). We use PointMetaBase-L as the source model and conduct experiment on 3DFRONT$\rightarrow$ S3DIS. From the results, we can see that due to the high complexity of the semantic segmentation model, updating too many parameters can actually lead to a decline in performance, as observed in the cases of AM and BC. In contrast, updating only a small number of important parameters, such as in our method or BN, can achieve better results.

Table 4: Ablation Study of the optimizable parameters

| | BN | AM | CH | EC | DC | Ours | BC |
|---|---|---|---|---|---|---|---|
| mIOU | 35.72 | 34.29 | 35.62 | 35.52 | 34.71 | 36.36 | 34.78 |

## A.4 CATEGORY-WISE RESULTS

### A.4.1 CATEGORY-WISE ENTROPY.

The class imbalance issue is very serious in point cloud scene segmentation, leading to significant differences in average entropy across different classes. To address this, we propose the Category-Balance Sampler. Here, we report the average entropy for each class for 3DFRONT$\rightarrow$S3DIS and 3DFRONT$\rightarrow$ScanNet, with the results shown in Table 5. It can be observed that minority classes typically have higher average entropy, which necessitates a higher entropy threshold to select more samples. This supports the validity of our Category-Balance Sampler.

### A.4.2 CATEGORY-WISE IoU.

To further explore how our method improves the source domain model, we report the IoU for 3DFRONT$\rightarrow$ ScanNet, with the results shown in Table 6. It can be observed that a significant improvement of NGTTA lies in its ability to enhance the performance of minority and difficult classes effectively. In contrast, TENT's inability to address the class imbalance issue results in a decline in the performance of minority classes, leading to training collapse.

Table 5: **Category-Wise Entropy Result on ScanNet and S3DIS.**

(a) ScanNet

|  | wall | floor | cabinet | bed | chair | sofa | table | door | window | bookshelf | desk |
|---|---|---|---|---|---|---|---|---|---|---|---|
| Quantity Ratio | 20% | 20% | 3% | 2% | 6% | 2% | 3% | 3% | 2% | 2% | 1% |
| Entropy | 0.40 | 0.50 | 0.70 | 0.66 | 0.57 | 0.56 | 0.58 | 0.53 | 0.93 | 0.88 | 0.98 |

(b) S3DIS

|  | wall | floor | chair | sofa | table | door | window | bookshelf | ceiling | beam | column |
|---|---|---|---|---|---|---|---|---|---|---|---|
| Quantity Ratio | 27% | 15% | 2% | 1% | 3% | 3% | 3% | 11% | 19% | 1% | 1% |
| Entropy | 0.14 | 0.19 | 0.71 | 0.41 | 0.89 | 0.59 | 0.50 | 0.22 | 0.15 | 0.60 | 0.58 |

Table 6: **Category-Wise IoU Result on 3DFRONT→ScanNet.**

| Method | wall | floor | cabinet | bed | chair | sofa | table | door | window | bookshelf | desk | mIoU |
|---|---|---|---|---|---|---|---|---|---|---|---|---|
| Baseline | 60.80 | 83.22 | 13.15 | 47.93 | 56.35 | 47.38 | 39.61 | 1.85 | 3.28 | 18.95 | 21.07 | 34.80 |
| TENT | 47.12 | 60.55 | 0.03 | 12.79 | 4.35 | 2.29 | 30.22 | 0 | 0 | 0 | 0 | 15.63 |
| **NGTTA** | 60.12 | 86.35 | 9.29 | 42.75 | 57.97 | 44.87 | 44.85 | 3.20 | 7.13 | 31.26 | 27.04 | **38.42** |

## A.5 ADDITIONAL EXPERIMENTS

### A.5.1 SemanticKITTI.

Here, we introduce a more challenging experiment by transferring from indoor data (3DFRONT→SemanticKITTI) to outdoor data to demonstrate the generalization capability of NGTTA. However, a significant challenge arises because indoor and outdoor datasets do not share the same classes, which prevents the use of common technical components such as pseudo-labeling, entropy minimization, and subsequent evaluation phases. Therefore, we only utilize a non-parametric geometric model for feature distillation, extracting point-level features from both the source model and the adaptive model, and we use SVM to evaluate accuracy. The results are shown in Table 7, where we can see that the accuracy significantly improves after using NGTTA. This demonstrates that NGTTA is also applicable to outdoor data and can enhance feature distinguishability.

Table 7: **SVM Accuracy of NGTTA on 3DFRONT→SemanticKITTI.**

|  | PointMetaBase-L | **+NGTTA** |
|---|---|---|
| Accuracy | 30.5 | 35.1 |

### A.5.2 COMPARE TO FPFH.

Here, we compare the performance of the geometric feature FPFH and PointNN on 3DFRONT→ScanNet and 3DFRONT→S3DIS to demonstrate that, in our method, PointNN is a superior non-parametric geometric feature extractor. The results are shown in Table 8. It can be observed that PointNN outperforms FPFH. We believe this is primarily because FPFH calculates geometric features based solely on the local neighborhood of each point, resulting in a very limited receptive field. In contrast, PointNN expands the receptive field by aggregating geometric features through multiple layers of down-sampling and up-sampling, which is crucial for scene segmentation.

Table 8: **Results of FPFH.**

|  | 3DFRONT→S3DIS | 3DFRONT→ScanNet |
|---|---|---|
| PointMetaBase-L | 29.38 | 34.80 |
| +FPFH | 34.98 | 36.76 |
| **+PointNN** | **36.36** | **38.42** |

### A.5.3 RESULTS ON ADDITIONAL MODELS.

in this part, we have added experiments with additional models MinkowskiNet (ResNet-UNet) and RandLA-Net on 3DFRONT→S3DIS and 3DFRONT→ScanNet, and the results are shown in Table 9. As can be seen, NGTTA can be applied to various point cloud architectures, demonstrating the generalizability of our method.

Table 9: **Results of MinkowskiNet and RandLA-Net.**

|  | 3DFRONT→S3DIS | 3DFRONT→ScanNet |
|---|---|---|
| MinkowskiNet | 24.35 | 31.72 |
| **+NGTTA** | 32.57 | 35.76 |
| RandLA-Net | 25.81 | 32.16 |
| **+NGTTA** | 33.71 | 35.91 |

## A.6 VISUALIZATION

### A.6.1 SEGMENTATION RESULTS

In this part, we compare the visualization results of the Source Model and the results after applying NGTTA adaptation, as shown in Figure 5. It can be seen that NGTTA effectively corrects the erroneous class predictions of the Source Model, resulting in improved segmentation results.

### A.6.2 ADAPTION PROCESS

Here, we visualize the domain adaptation process using NGTTA, with the results shown in Figure 6. (a) represents the feature distribution obtained by directly applying the source model on the target data, while (d) represents the feature distribution of the model trained on labeled data in the target domain, which can be considered as the target domain distribution. (b) and (c) show the results after training with NGTTA for 1 and 2 epochs, respectively. We can see that the black and red circles highlight the areas where the source domain distribution and the target domain distribution differ significantly. As the NGTTA training progresses, the model's feature distribution gradually approaches the target domain distribution.

## A.7 INTRODUCTION OF POINTNN

PointNN is a non-parametric geometric model that utilizes common components from point cloud models, such as Farthest Point Sampling (FPS), kNN, and max pooling, to extract local features from point clouds. Specifically, for the $i$-th point $p_i = (x_i, y_i, z_i) \in R^{1\times3}$ in the point cloud, it first employs trigonometric functions to extract positional features.

$$f_i^x = [sine(Ax_i/B^{\frac{6\cdot0}{C}}), Cosine(Ax_i/B^{\frac{6\cdot0}{C}}), ..., sine(Ax_i/B^{\frac{6\cdot m}{C}}), Cosine(Ax_i/B^{\frac{6\cdot m}{C}})] \quad (2)$$

where $f_i^x$ is the positional encoding for the x-axis. $A$ and $B$ means the magnitude and wavelengths. The encoding for the y-axis and z-axis is the same. Therefore, the position embedding of $p_i$ can be written as:

$$f_i = PoE(p_i) = [f_i^x, f_i^y, f_i^z] \quad (3)$$

Then, PointNN use kNN to construct the neighborhood $(p_j, f_j)_{j\in\mathcal{N}_i}$ of the $i$-th center point. Then, for each neighborhood vector, PointNN expands it to:

$$f_{ij} = [f_i, f_j] \quad (4)$$

To capture the relevant geometric information between the center point and the neighboring points, PointNN incorporates relative positional encoding:

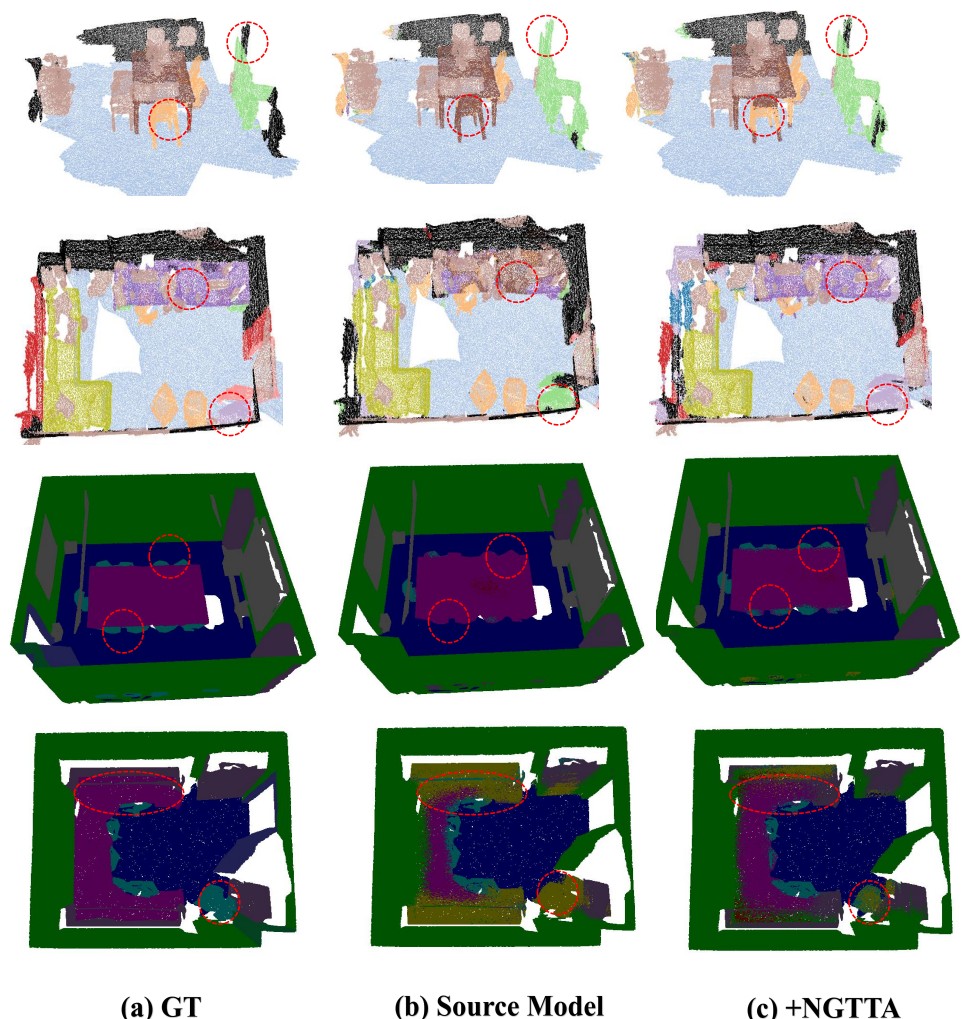

|  (a) GT | (b) Source Model | (c) +NGTTA |

Figure 5: Visualization comparison of NGTTA and Source Model. The first column is the label, the second column is the segmentation result from the Source Model, and the last column is the segmentation result after applying NGTTA adaptation. Then, the first two rows are the results for 3DFRONT→ScanNet, and the last two rows are the results for 3DFRONT→S3DIS.

$$\hat{f}_{ij} = (f_{ij} + PoE((p_i - p_j))) \odot PoE((p_i - p_j)) \tag{5}$$

Finally, PointNN uses max pooling and avg pooling to aggregate the local features.

$$\hat{f}_i = MaxPool(\{\hat{f_{ij}}\}_{j \in \mathcal{N}_i}) + AvgPool(\{\hat{f_{ij}}\}_{j \in \mathcal{N}_i}) \tag{6}$$

PointNN extracts rich geometric information from local regions by employing multi-layer down-sampling and aggregating local features and then obtains point-wise features through multi-layer upsampling.

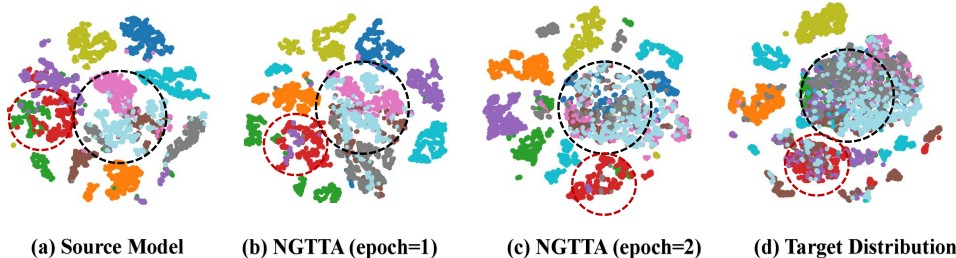

(a) Source Model     (b) NGTTA (epoch=1)     (c) NGTTA (epoch=2)     (d) Target Distribution

Figure 6: TSNE of NGTTA Results on 3DFRONT→S3DIS. The **black** and **red** circles represent the parts where the source model distribution and the target domain distribution differ significantly. It can be observed that as NGTTA continues to train, the model's feature distribution gradually approaches the target domain distribution.

