# OpenReview forum: "NGTTA: Non-parametric Geometry-driven Test Time Adaption for 3D Point Cloud Segmentation"
_ICLR.cc/2025/Conference — Submitted to ICLR 2025_

### Official Review · Reviewer_86r6 · 2024-10-29

**Soundness:** 2
**Presentation:** 2
**Contribution:** 3
**Rating:** 6
**Confidence:** 3

**Summary:**

This paper addresses the problem of the domain gap between source and target data for the task of 3D point cloud segmentation. The proposed NGTTA is a Test Time Adaptation (TTA) method that leverages a non-parametric geometric model. This model is used for producing a so-called intermediate domain, guided by the intuition that the geometric features in the source data and target data could be utilized to bridge the domain gap. Furthermore, the introduced methodology addresses the problem of category imbalance by a category-balance sampler filtering easy and hard samples. The evaluation on the sim-to-real benchmarks shows that the proposed method performs better than the competitors.

**Strengths:**

1. The method introduced by the authors addresses the challenging problem of the domain shift for the task of 3D semantic segmentation and shows that geometric priors could bridge this gap.
2. Various experimental setups and ablations (seem to) speak to the effectiveness of the proposed approach (however, the details should be clarified — see Questions). The reporting of the computational overhead introduced by the method is also appreciated.

**Weaknesses:**

1. One drawback of the paper is the lack of clarity in writing. The paper could significantly benefit from polishing the text. A *non-exhaustive* list of issues is given in the questions.
2. The prior work PointNN by Zhang et al. (2023) appears to play the central part of the proposed method, since it is the non-parametric geometric model; yet, it needs to be recapped sufficiently in the main part of the paper or in the appendix.
3. The hyperparameter selection is biased to the target domain:
Many components of the proposed method rely on hyperparameters, e.g., $\sigma$ and $\gamma$ in the sampling (4), $\alpha_i$ in the total loss (A.2 eq. (1) ), the $k$ in kNN, as shown by the deviation of the results in the respective tables.
The authors use the 3DFRONT→S3DIS benchmark to find the hyperparameters.
Question: Were these selected hyperparameters also used for the 3DFRONT→ScanNet benchmark?
(See also Q2 in Quesitions.)
If so, in the best case, there’s only one true setup, where the target data is not accessed to modify the method — 3DFRONT→ScanNet.

**Questions:**

1. Could the authors address the following:
- “non-parametric geometric” (e.g., L032, L069, L098, L157, and others)— is “non-parametric geometry” meant or “non-parametric geometric model”?
- grammar in L070-071?
- “data of geometric.” in L156?
- L263, “source data” —> “target data”.
- the citations need to be adjusted — no need for parentheses when in text, only a single pair of parentheses otherwise.
2. In Tables 1-3 in the Appendix, there are $4^3 = 64$ different combinations of $\alpha_1$, $\alpha_2$, and $\alpha_3$.
- Which 12 combinations are presented in the tables?
3. Does the *j*-th sample in (5) mean a *j*-th point in the *i*-th point cloud (scene)?
4. How is $\beta$ in (11) chosen in practice?
5. Did the authors perform the comparison only for a single initialization? I.e., only one test run?
6. Does the non-parametric model have a decoder as visualized in Figure 2?
7. Comment: The arrow between $\Gamma^{ps}$ and $L_{cls}$ n Figure 2 should be fixed.
8. Comment: The paper would benefit from having additional qualitative results, especially for the 3DFRONT→ScanNet  benchmark, for which they are currently missing.

---

> ### Author Response · Authors · 2024-11-20
>
> **Q1:** One drawback of the paper is the lack of clarity in writing. The paper could significantly benefit from polishing the text. A non-exhaustive list of issues is given in the questions.
>
> **R1:** Thank you for your suggestions. We have corrected these errors and carefully review the writing in the revision of our paper. The following are responses to some writing inquiries:
>
> 1.1 "non-parametric geometric" refers to non-parametric geometric features, specifically the point-wise features extracted using PointNN in our method.
>
> 1.2 “data of geometric.” This is a writing error, and we will make corrections in the next version.
>
> 1.3 "source data in L263."
> This statement is correct; it means that the previous method requires obtaining source domain data and target domain data in order to customize the "intermediate domain".
>
> **Q2:**  The prior work PointNN by Zhang et al. (2023) appears to play the central part of the proposed method, since it is the non-parametric geometric model; yet, it needs to be recapped sufficiently in the main part of the paper or in the appendix.
>
> **R2:** Thank you for your suggestions. In the uploaded revised version of our paper, we have included the introduction of PointNN \textbf{in Appendix A.7}.
>
> **Q3:** Which 12 combinations of $\alpha$ are presented in the tables?
>
> **R3:** Sorry for the confusion. In our experiments, $\alpha_1$=10, $\alpha_2$=0.5, and $\alpha_3$=1 yield the best performance. Therefore, when conducting the ablation study for $\alpha_1$, we fixed $\alpha_2$ at 0.5 and $\alpha_3$ at 1. The same approach applies for the ablation studies of $\alpha_2$ and $\alpha_3$ as well.
>
> **Q4:** The hyperparameter selection is biased to the target domain...
>
> **R4:** Thank you for your question. In our setup, apart from $\sigma$ and $\gamma$, the other hyperparameters are universally applicable across all datasets, as their adjustments need to be selected based on final performance. To ensure the applicability of our method, we conducted ablation experiments on 3DFRONT$\rightarrow$S3DIS to select the other hyperparameters, keeping them constant across all datasets.
>
> As for $\sigma$ and $\gamma$, they are primarily chosen based on the average entropy of the datasets, since the entropy distributions vary across different datasets. For instance, the average entropy for ScanNet is 0.6, while for S3DIS it is 0.3. We believe this approach is acceptable because the average entropy can be obtained with a single forward pass of the model on the target domain without requiring additional training.
>
> Certainly, in future work, we will explore more universal hyperparameter settings to enhance the generalizability of our method.
>
> **Q5:** Does the $j$-th sample in (5) mean a $j$-th point in the $i$-th point cloud (scene)?
>
> **R5:** Yes, in Eq (5), $E_{ij}$ represents the entropy of the $j$-th point in the $i$-th scene.
>
> **Q6:** How is
>  in (11) chosen in practice?
>
> **R6:** Thank you for your question. The $\beta$ is set to 0.9 to emphasize non-parametric geometric features more in the early stages of training, while focusing more on well-adapted model features in the later stages of training.
>
> **Q7:**  Did the authors perform the comparison only for a single initialization? I.e., only one test run?
>
> **R7:**  In our method comparison, the TTA series of methods, including NGTTA, are all trained on pre-trained source models, ensuring that the initialization weights are consistent. Additionally, we do not employ data augmentation techniques, which results in minimal performance variability, making the comparison fair.
>
> **Q8:** Does the non-parametric model have a decoder as visualized in Figure 2
>
> **R8:** Yes, PointNN extracts point-wise non-parametric geometric features through multi-layer downsampling (encoder) and multi-layer upsampling (decoder).
>
> **Q9:** The arrow in Figure 2 should be fixed.
>
> **R9:** Thank you for your careful review. **We have made modifications to Figure (2) in the revised version.**
>
> **Q10:** The paper would benefit from having additional qualitative results, especially for the 3DFRONT→ScanNet benchmark, for which they are currently missing.
>
> **R10:**  Thank you for your suggestions. In the uploaded revised version of our paper, we have included the visualization comparison of NGTTA and Source Model on 3DFRONT$\rightarrow$S3DIS and 3DFRONT$\rightarrow$ScanNet **in Appendix A.6.1**  and the visualization of TSNE during the adaptation process **in Appendix A.6.2**.  Through these visualizations, we can gain a clear understanding of the NGTTA adaptation process and the superiority of the results.

---

> ### Author Response · Authors · 2024-11-25
> **End of Discussion Approaching**
>
> Dear Reviewer 86r6:
>
> We sincerely thank you for taking the time to provide detailed comments. We have made corresponding revisions based on your questions and added a substantial amount of experiments.
>
> Given that the end of the discussion period is approaching, we would like to ask if there are any unclear aspects regarding our work. We are more than happy to provide further explanations as needed.
>
> Our work primarily explores Test Time Adaptation in point cloud scenes, conducted under the challenging SIm2Real setting, which is still an underexplored area. Our approach involves innovative attempts and achieves state-of-the-art performance. We hope that this work will be recognized and valued.
>
> We would like to express our gratitude once again for your feedback and suggestions, which have greatly helped us improve our work.
>
> Best and sincere wishes,
>
> The authors

---

> > ### Comment · Reviewer_86r6 · 2024-11-25
> > **Updated assessment**
> >
> > As the authors addressed the reviewer's concerns, the assessment was updated. The reviewer would still like to see further explanations and descriptions, in particular of the hyperparameter selection and the ablation.

---

> > > ### Author Response · Authors · 2024-11-25
> > > **Thank You Vey Much for Your Response**
> > >
> > > Dear Reviewer 86r6:
> > >
> > > Thank you very much for the follow-up! We will conduct more ablation experiments in the next version.
> > >
> > > Thank you once again for your feedback and suggestions. Best and sincere wishes,
> > >
> > > The authors

---

### Official Review · Reviewer_dGga · 2024-10-29

**Soundness:** 2
**Presentation:** 2
**Contribution:** 3
**Rating:** 5
**Confidence:** 4

**Summary:**

This paper proposes NGTTA, a method for test-time adaptation in 3D point cloud segmentation. NGTTA leverages the distribution of non-parametric geometry as an “intermediate domain” to facilitate model adaptation. The approach involves using a source model alongside a non-parametric geometric model to extract features from point clouds. To handle class imbalance, a category-balance sampler separates easy and hard samples. For easy samples, pseudo-labels and entropy minimization are applied, with soft voting used for correction among neighboring points. Hard samples, on the other hand, are distilled into the source model, allowing the model to quickly converge to the intermediate domain that resembles the target.

**Strengths:**

- The proposed model shows meaningful performance gain for sim-to-real and cross-domain scenarios.

- Extensive experiments and analysis are provided in the paper.

**Weaknesses:**

- It seems that the explanation regarding the motivation for using the non-parametric geometric model for domain generalizability is insufficient. Is there a way to demonstrate that the non-parametric geometric model is generalizable?

- If I understand correctly, the non-parametric geometric model ultimately allows for good clustering without the need for learning, but I’m not sure if applying it at the feature level is the best approach. If the non-parametric model is truly domain generalizable, wouldn’t it be a better approach to use it simply as a refinement method in the target domain as a form of post-processing, or to utilize it when generating pseudo-GT in a pseudo-labeling approach?

- The paper lacks sufficient qualitative results, which makes it difficult to fully understand or visualize the practical effectiveness of the proposed method.

- There are writing errors, such as the sudden introduction of "S2T" in the caption of Figure 1 without prior explanation.

**Questions:**

- Could you clarify which point in time the features in Figure 3 represent? Additionally, while (b) seems to demonstrate promising results, there is no accompanying explanation. Could you provide further details regarding this?

- Is this methodology applicable to LiDAR data as well?

---

> ### Author Response · Authors · 2024-11-20
>
> **Q1:** It seems that the explanation regarding the motivation for using the non-parametric geometric model for domain generalizability is insufficient. Is there a way to demonstrate that the non-parametric geometric model is generalizable?
>
> **R1:** hank you for your suggestion. The inspiration for this idea mainly stems from the results of several papers. Firstly, regarding PointNN [1], the results in its original paper demonstrate that PointNN approaches the performance of fully supervised models like PointNet and PointNet++ even without parameters. Additionally, PointNN outperforms methods such as PointCLIP, PointCLIPV2, and ULIP in the absence of any supervisory signals, indicating that PointNN is an effective feature extractor for unlabeled new data. Furthermore, explicit geometric information has been shown to possess strong generalization capabilities, as evidenced by studies like X-3D [2] and RepSurf [4], which validate this conclusion. Moreover, PointDGMamba [3] has demonstrated in experiments that incorporating explicit geometric information like X-3D can significantly enhance domain generalization capabilities. Therefore, these results suggest that non-parametric geometric models should be effective in domain generalization tasks.
>
>
> [1] Zhang R, Wang L, Guo Z, et al. Parameter is not all you need: Starting from non-parametric networks for 3d point cloud analysis[J]. arXiv preprint arXiv:2303.08134, 2023.
>
> [2] Sun S, Rao Y, Lu J, et al. X-3D: Explicit 3D Structure Modeling for Point Cloud Recognition[C]//Proceedings of the IEEE/CVF Conference on Computer Vision and Pattern Recognition. 2024: 5074-5083.
>
> [3] Yang H, Zhou Q, Sun H, et al. PointDGMamba: Domain Generalization of Point Cloud Classification via Generalized State Space Model[J]. arXiv preprint arXiv:2408.13574, 2024.
>
> [4] Ran H, Liu J, Wang C. Surface representation for point clouds[C]//Proceedings of the IEEE/CVF conference on computer vision and pattern recognition. 2022: 18942-18952.
>
>
> **Q2:** If I understand correctly, the non-parametric geometric model ultimately allows for good clustering without the need for learning, but I’m not sure if applying it at the feature level is the best approach. If the non-parametric model is truly domain generalizable, wouldn’t it be a better approach to use it simply as a refinement method in the target domain as a form of post-processing, or to utilize it when generating pseudo-GT in a pseudo-labeling approach?
>
> **R2:** Yes, your understanding is correct. We do apply the non-parametric geometric features to the generation of pseudo-labels, and you can refer to Eq. (10) in the paper for details. However, we believe that due to the challenging task of scene segmentation, the pseudo-labels of many difficult samples are inaccurate, and even label correction by non-parametric geometric features is difficult to solve. Therefore, it is not enough to only use non-parametric geometric features to generate pseudo-labels. To this end, we additionally introduce the distillation training of geometric features for difficult samples to improve the performance of scene segmentation tasks.
>
> **Q3:** The paper lacks sufficient qualitative results, which makes it difficult to fully understand or visualize the practical effectiveness of the proposed method.
>
> **R3:** Thank you for your suggestions.
> In the uploaded revised version of our paper, we have included the visualization comparison of NGTTA and Source Model on 3DFRONT$\rightarrow$S3DIS and 3DFRONT$\rightarrow$ScanNet **in Appendix A.6.1**  and the visualization of TSNE during the adaptation process **in Appendix A.6.2**.  Through these visualizations, we can gain a clear understanding of the NGTTA adaptation process and the superiority of the results.
>
> **Q4:** There are writing errors, such as the sudden introduction of "S2T" in the caption of Figure 1 without prior explanation.
>
> **R4:** Sorry for the confusion. This was our oversight. "S2T" refers to the feature distribution of the source model in the target domain, and **we have updated Figure (1) in the revised version of the paper**.

---

> > ### Author Response · Authors · 2024-11-20
> >
> > **Q5:** Could you clarify which point in time the features in Figure 3 represent? Additionally, while (b) seems to demonstrate promising results, there is no accompanying explanation. Could you provide further details regarding this?
> >
> > **R5:** Sorry for the confusion. First, the orange section in Figure (3) refers to the feature distribution trained on the target domain using labels. In (a), it represents the feature distribution of the source model when it is directly inferred on the target domain without adaptation. (b) indicates the feature distribution obtained after adapting the source model using AdaContrast. (c) refers to the feature distribution of the non-parametric geometric model in the target domain, while (d) shows the feature distribution obtained after adapting the source model using NGTTA.
> >
> > Regarding (b), it is true that AdaContrast previously achieved the best performance among previous TTA methods due to the incorporation of more complex components like contrastive learning. However, there are two issues: first, it has a very high computational cost due to the addition of contrastive loss, which is challenging for scene point clouds. Second, compared to (c), its feature distribution still shows some differences from the target domain distribution, resulting in performance that is still inferior to NGTTA.
> >
> > **Q6:** Is this methodology applicable to LiDAR data as well?
> >
> > **R6:**  Thank you for your question. We have included experiments for adapting from indoor data to outdoor data, 3DFRONT$\rightarrow$ SemanticKITTI, which is shown in **Table R6**. However, due to the lack of shared categories, common techniques such as entropy minimization and pseudo-labeling, as well as precise category accuracy assessments, cannot be applied. To address this, we employed non-parametric geometric features for feature distillation to achieve domain adaptation. We output the point-wise features from both the source model and the adapted model, and evaluated the accuracy using an SVM classifier. This experiment allows us to determine whether NGTTA is suitable for LIDAR data to enhance its feature distinguishability. **In the uploaded revised version of our paper, we have included this additional result in Appendix A.5.1 .**
> >
> > **Table R6: SVM Accuracy of NGTTA on 3DFRONT $\rightarrow$SemanticKITTI**
> >
> > |               | PointMetaBase-L | **+NGTTA** |
> > |---------------|-----------------|------------|
> > | Accuracy     | 30.5            | 35.1       |

---

> ### Author Response · Authors · 2024-11-25
> **End of Discussion Approaching**
>
> Dear Reviewer Reviewer dGga:
>
> We sincerely thank you for taking the time to provide detailed comments. We have made corresponding revisions based on your questions and added a substantial amount of experiments.
>
> Given that the end of the discussion period is approaching, we would like to ask if there are any unclear aspects regarding our work. We are more than happy to provide further explanations as needed.
>
>
> Our work primarily explores Test Time Adaptation in point cloud scenes, conducted under the challenging SIm2Real setting, which is still an underexplored area. Our approach involves innovative attempts and achieves state-of-the-art performance. We hope that this work will be recognized and valued.
>
> We would like to express our gratitude once again for your feedback and suggestions, which have greatly helped us improve our work.
>
> Best and sincere wishes,
>
> The authors

---

> > ### Comment · Reviewer_dGga · 2024-11-27
> >
> > Thank you for your response as described above. However, I still have some concerns regarding certain parts, so I would like to maintain the rating.

---

> > > ### Author Response · Authors · 2024-11-27
> > >
> > > Thank you for your reply. We would like to ask what you are not clear and worried about so that we can further clarify.

---

> ### Comment · Reviewer_dGga · 2024-11-30
>
> I have some doubts regarding the motivation for using a non-parameterized model and whether the proposed method for its usage is optimal. Additionally, I find it difficult to fully accept the academic contribution claimed in the paper.

---

> > ### Author Response · Authors · 2024-11-30
> >
> > Dear Reviewer dGga:
> >
> > Thank you for your response. I hope the following clarifications will address your concerns:
> >
> > 1. **As we have emphasized multiple times, explicit geometric information has been shown to effectively serve as a data prior or regularization method in many other advanced works**, providing insights into the distribution of the dataset. Therefore, in our approach, the explicit geometric information extracted from the non-parametric geometric model is used as a regularization mechanism during the early stages of self-training to stabilize the training process, gradually learning the distribution information of the dataset for transfer to the target domain. We believe this motivation is both reasonable and clear.
> > Additionally, our experimental results indicate that simply introducing features extracted from the non-parametric geometric model can improve the mIoU by approximately 2\%, further validating the rationale of applying non-parametric geometric models to domain adaption.
> >
> > 2. To the best of our knowledge, we are the first to apply Test Time Adaptation (TTA) to indoor scene segmentation. We compared our method with state-of-the-art 2D TTA methods, such as AdaContrast, and achieved significant performance improvements, increasing the model's performance by over 3\%. This indicates that our approach is highly superior.
> >
> > 3. Our academic contributions can be summarized in three main points:
> >
> > &ensp; &ensp; &ensp; (a) We are the first to apply Test Time Adaptation (TTA) to 3D indoor scene segmentation, and our experiments demonstrate that previous 2D TTA methods typically perform poorly when applied to 3D indoor scene segmentation.
> >
> > &ensp; &ensp; &ensp; (b) We discovered that non-parametric geometric models have strong generalization capabilities and can be applied to any target domain as a regularization mechanism to stabilize the domain adaptation process of the source model. This provides a fresh perspective for applying TTA methods to point cloud tasks.
> >
> > &ensp; &ensp; &ensp; (c) We identified that one factor limiting the performance of TTA in scene segmentation is class imbalance. To address this issue, we proposed a simple entropy-based class balancing sampling mechanism that effectively resolves the class imbalance problem. Experimental results also show that our method performs well on minority classes.
> >
> >
> > We sincerely hope that you will reconsider our contributions.
> >
> > Best and sincere wishes,
> >
> > The authors

---

### Official Review · Reviewer_71ZT · 2024-11-03

**Soundness:** 3
**Presentation:** 3
**Contribution:** 3
**Rating:** 6
**Confidence:** 3

**Summary:**

This paper introduces a Test-Time Adaptation (TTA) method for point cloud segmentation on indoor datasets. The method utilizes easy samples to minimize entropy loss and employs pseudo-label predictions to fine-tune the source domain model by considering both semantic and geometric k-nearest neighbors (non-parametric PointNN). By leveraging the ability of non-parametric geometry to represent the underlying manifolds of the target data, this approach effectively reduces the challenges associated with domain adaptation. Experimental results demonstrate improved performance on 3D segmentation tasks.

**Strengths:**

The paper is well-structured and provides a robust methodology that is rigorously evaluated on challenging sim-to-real benchmarks (e.g., 3DFRONT→ScanNet and 3DFRONT→S3DIS). The experimental results demonstrate a good improvement on indoor datasets. The proposed category-balance sampler and soft voting mechanisms are well-motivated and backed by quantitative results. Additionally, the authors conducted thorough ablation studies to evaluate different components of the framework, such as the impact of soft voting, entropy threshold, and loss weights.

**Weaknesses:**

The technical contribution of this paper appears relatively weak, as several components are directly adapted from existing 2D TTA methods. For instance, entropy selection and moment update mechanisms are borrowed directly from established 2D TTA techniques [1] without significant innovation in the 3D domain (only add a hard samples selection). Additionally, components such as entropy minimization and pseudo-labeling have been widely explored in related fields, including 3D semi-supervised segmentation and unsupervised learning[3]. The authors could strengthen the paper by either justifying the unique adaptations of these methods to 3D point cloud segmentation or by exploring alternative techniques tailored to the specific challenges of the 3D domain.
Furthermore, it would be valuable to discuss why other geometric features commonly used in 3D point cloud analysis, such as Fast Point Feature Histograms (FPFH), were not considered. Comparing the performance of the non-parametric geometric approach against more established geometric features would provide stronger evidence of its efficacy and clarify the specific benefits of the proposed method.

[1] Wang, Dequan, et al. “Tent: Fully test-time adaptation by entropy minimization.” International Conference on Learning Representations (ICLR). 2021.
[2] Xie, Qizhe, et al. “Self-training with noisy student improves imagenet classification.” IEEE/CVF Conference on Computer Vision and Pattern Recognition (CVPR). 2020.
[3] Hou, Yikang, et al. “3D-SST: Self-Supervised Pretraining for 3D Scene Understanding.” European Conference on Computer Vision (ECCV). 2022.

**Questions:**

1. The statement "We can see that as the sample count decreases, the threshold will gradually increase" is reasonable. However, it’s stated that tail categories with fewer samples will have a larger entropy threshold and "We can see that as the sample count decreases, the threshold will gradually increase", but the reasoning behind this isn't obvious from the equation itself.

2. Whether Eq.~(4) is applied individually to each point cloud or collectively across all point clouds. Does each point cloud have its own unique $\sigma_c$, or do all point clouds share the same $\sigma_c$?

3. For Eq.~(13), the notation could be confusing for readers. Since $\mathbf{f}_i$ represents a vector, and $\mathbf{f}_{ij}$ also appears to be a vector, the mathematical symbols could be refined to avoid potential misinterpretation.

4. Lastly, it would be insightful to discuss the model’s performance when transitioning from indoor datasets to outdoor datasets. Are there any notable differences in performance, or specific challenges encountered when adapting to outdoor data?

5. Although the method’s applicability to multiple architectures is partially demonstrated, more rigorous evaluation on additional 3D segmentation backbones (e.g., MinkowskiNet, RandLA-Net) would help generalize the findings. Providing results for diverse architectures could highlight the robustness of the NGTTA framework, particularly if model performance varies significantly with architecture.

6. Visualization of Intermediate Domain Effects: While the paper presents the concept of an “intermediate domain,” it would benefit from visualizations that illustrate this intermediate representation’s impact on the distribution alignment between source and target domains. For example, t-SNE or UMAP plots showing the evolution of feature space alignment as the model adapts would provide more concrete evidence of the intermediate domain’s role in reducing domain gaps.

7. Comparing the performance of the non-parametric geometric approach against more established geometric features such as FPFH would provide stronger evidence of its efficacy and clarify the specific benefits of the proposed method.

---

> ### Author Response · Authors · 2024-11-20
>
> **Q1:** The statement "We can see that as the sample count decreases, the threshold will gradually increase" is reasonable. However, it’s stated that tail categories with fewer samples will have a larger entropy threshold and "We can see that as the sample count decreases, the threshold will gradually increase", but the reasoning behind this isn't obvious from the equation itself.
>
> **R1:** Sorry for the confusion. First, the statement "We can see that as the sample count decreases, the threshold will gradually increase" mainly refers to Eq. (4), which indicates that in our sampling mechanism, the threshold is inversely proportional to the number of classes. This demonstrates that our sampling mechanism allows for a higher threshold for minority classes to sample more instances.
>
> As for the assertion that minority classes typically have higher entropy, **this conclusion is drawn from observations of experimental results rather than derived from the formula**. As shown in  **Table R1**, we report the proportions and average entropy for each class in ScanNet and S3DIS.
> We can see that the entropy of minority classes is typically much higher than that of majority classes, which supports the conclusion above. **In the uploaded revised version of our paper, we have included this additional result in Appendix A.4.1**
>
> **Table R1: Category-Wise Entropy Result on ScanNet and S3DIS**
>
> ### (a) ScanNet
>
> |          | wall | floor | cabinet | bed | chair | sofa | table | door | window | bookshelf | desk |
> |----------|------|-------|---------|-----|-------|------|-------|------|--------|-----------|------|
> | Quantity Ratio | 20%  | 20%   | 3%      | 2%  | 6%    | 2%   | 3%    | 3%   | 2%     | 2%        | 1%   |
> | Entropy  | 0.40 | 0.50  | 0.70    | 0.66| 0.57  | 0.56 | 0.58  | 0.53 | 0.93   | 0.88      | 0.98 |
>
> ### (b) S3DIS
>
> |          | wall | floor | chair | sofa | table | door | window | bookshelf | ceiling | beam | column |
> |----------|------|-------|-------|------|-------|------|--------|-----------|---------|------|--------|
> | Quantity Ratio | 27%  | 15%   | 2%    | 1%   | 3%    | 3%   | 3%     | 11%       | 19%     | 1%   | 1%     |
> | Entropy  | 0.14 | 0.19  | 0.71  | 0.41 | 0.89  | 0.59 | 0.50   | 0.22      | 0.15    | 0.60 | 0.58   |
>
> **Q2:** Whether Eq.~(4) is applied individually to each point cloud or collectively across all point clouds. Does each point cloud have its own unique $\sigma_c$
> , or do all point clouds share the same $\sigma_c$ ?
>
> **R2:** Sorry for the confusion. $\sigma_c$ is shared among all point clouds in the dataset.
>
> **Q3:** For Eq.~(13), the notation could be confusing for readers. Since $\mathbf{f}i
> \mathbf{f}{ij}$ also appears to be a vector, the mathematical symbols could be refined to avoid potential misinterpretation
>
> **R3:** Sorry for the confusion.  $f_i\in R^{M\times C}$ represents the features of the $i-th$ scene, which is defined in Eq. (1). $f_{ij}$ represents the features of the $j-th$ point in the $i-th$ scene, as defined in line **284**. We will emphasize these definitions again before Eq. (13) in the next version.
>
> **Q4:** Lastly, it would be insightful to discuss the model’s performance when transitioning from indoor datasets to outdoor datasets. Are there any notable differences in performance, or specific challenges encountered when adapting to outdoor data?
>
> **R4:** Thank you for your advice. Firstly, Test Time Adaption and domain transfer usually difficult to transfer models from indoor datasets to outdoor datasets, because their categories are not shared, which makes techniques such as pseudo-label, entropy minimization and subsequent evaluation stages impossible. To prove that NGTTA is also applicable to outdoor data, we do a simple experiment where we take the source  model PointMetaBase-L trained on **3DFRONT**, distill it using a non-parametric geometric model, and update the BatchNorm parameter of the model on the **SemanticKITTI** dataset. Then we extract the point-wise features of the source  model and the updated model respectively, and judge the discriminability of the features by SVM classifier. We hope to use this simple experiment to judge whether NGTTA can improve the feature representation ability of the source  model on the outdoor data set, and the results are shown in **Table R4**. It can be seen that by introducing a non-parametric geometric model, the feature discriminability of the model on outdoor data can be improved, which can prove the effectiveness of NGTTA to some extent.  **In the uploaded revised version of our paper, we have included this additional experiment in Appendix A.5.1 .**
>
> **Table R4: SVM Accuracy of NGTTA on 3DFRONT$\rightarrow$SemanticKITTI**
>
> |               | PointMetaBase-L | **+NGTTA** |
> |---------------|-----------------|------------|
> | Accuracy     | 30.5            | 35.1       |

---

> ### Author Response · Authors · 2024-11-20
>
> **Q5:** Although the method’s applicability to multiple architectures is partially demonstrated, more rigorous evaluation on additional 3D segmentation backbones (e.g., MinkowskiNet, RandLA-Net) would help generalize the findings. Providing results for diverse architectures could highlight the robustness of the NGTTA framework, particularly if model performance varies significantly with architecture.
>
> **R5:** Thank you for your suggestion. We have added experiments with MinkowskiNet (ResNet-UNet) and RandLA-Net on 3DFRONT$\rightarrow$S3DIS and 3DFRONT$\rightarrow$ScanNet, and the results are shown in **Table R5**. As can be seen, NGTTA can be applied to various point cloud architectures, demonstrating the generalizability of our method. **In the uploaded revised version of our paper, we have included this additional experiment in Appendix A.5.3 .**
>
> **Table R5: Results of MinkowskiNet and RandLA-Net**
>
> |                   | 3DFRONT → S3DIS | 3DFRONT → ScanNet |
> |-------------------|-----------------|-------------------|
> | MinkowskiNet      | 24.35           | 31.72             |
> | **+NGTTA**        | 32.57           | 35.76             |
> | RandLA-Net        | 25.81           | 32.16             |
> | **+NGTTA**        | 33.71           | 35.91             |
>
> **Q6:** Visualization of Intermediate Domain Effects: While the paper presents the concept of an “intermediate domain,” it would benefit from visualizations that illustrate this intermediate representation’s impact on the distribution alignment between source and target domains. For example, t-SNE or UMAP plots showing the evolution of feature space alignment as the model adapts would provide more concrete evidence of the intermediate domain’s role in reducing domain gaps.
>
> **R6:** Thank you for your suggestion. In fact, we attempted to visualize this concept in Figure (3) of the paper, but due to space limitations, we did not provide a detailed explanation. Here, we would like to add some clarification. Firstly, the orange section represents the feature distribution of the target domain, and we can see that the height of the feature distribution is quite consistent, indicating that sufficient supervisory signals have led to well-normalized features. Figure (3) (a) represents the feature distribution of the source model, which shows a highly inconsistent feature distribution, highlighting the differences in distribution. Figure (3) (c) displays the feature distribution of the non-parametric geometric model; although there are still some differences compared to the target domain distribution, the height of the feature distribution is also quite consistent. Compared to the source model, it is closer to the target domain distribution, suggesting that the non-parametric geometric model can be considered an intermediate domain.
>
> However, we believe your suggestion is better, as using t-SNE can provide a clearer visualization of feature alignment during the adaptation process. Therefore, **in the uploaded revised version of our paper, we have included the visualization of TSNE during the adaptation process in Appendix A.6.2 .**.  Furthermore, from the t-SNE visualization results, it can be seen that NGTTA effectively facilitates the gradual migration of the source model to the target domain distribution.
>
> **Q7:** Comparing the performance of the non-parametric geometric approach against more established geometric features such as FPFH would provide stronger evidence of its efficacy and clarify the specific benefits of the proposed method.
>
> **R7:** Thank you for your suggestion. We report the performance of FPFH and our chosen non-parametric geometric model, PointNN, on 3DFRONT$\rightarrow$S3DIS and 3DFRONT$\rightarrow$ScanNet, with the results shown in **Table R7**. It can be observed that PointNN outperforms FPFH. We believe this is primarily because FPFH calculates geometric features based solely on the local neighborhood of each point, resulting in a very limited receptive field. In contrast, PointNN expands the receptive field by aggregating geometric features through multiple layers of down-sampling and up-sampling, which is crucial for scene segmentation. **In the uploaded revised version of our paper, we have included this additional result in Appendix A.5.2 .**
>
> **Table R7: Results of FPFH**
> |               | 3DFRONT → S3DIS | 3DFRONT → ScanNet |
> |---------------|-----------------|-------------------|
> | PointMetaBase-L | 29.38          | 34.80             |
> | +FPFH          | 34.98           | 36.76             |
> | **+PointNN**   | **36.36**       | **38.42**         |

---

> ### Author Response · Authors · 2024-11-25
> **End of Discussion Approaching**
>
> Dear Reviewer 71ZT:
>
> We sincerely thank you for taking the time to provide detailed comments. We have made corresponding revisions based on your questions and added a substantial amount of experiments.
>
> Given that the end of the discussion period is approaching, we would like to ask if there are any unclear aspects regarding our work. We are more than happy to provide further explanations as needed.
>
>
> Our work primarily explores Test Time Adaptation in point cloud scenes, conducted under the challenging SIm2Real setting, which is still an underexplored area. Our approach involves innovative attempts and achieves state-of-the-art performance. We hope that this work will be recognized and valued.
>
> We would like to express our gratitude once again for your feedback and suggestions, which have greatly helped us improve our work.
>
> Best and sincere wishes,
>
> The authors

---

### Official Review · Reviewer_HLQ7 · 2024-11-04

**Soundness:** 3
**Presentation:** 3
**Contribution:** 3
**Rating:** 5
**Confidence:** 3

**Summary:**

This paper proposes a non-parametric geometry-based test time adaption approach, named NGTTA,  to improve the domain adaption ability for 3D point cloud semantic segmentation. NGTTA mainly introduces a no-parametric model to extract features as intermediate domain and proposes a category-balance sampler to address class imbalance. Evaluations on 3DFRONT to ScanNet and D3DIS settings show the practicality of the proposed method.

**Strengths:**

- The motivation is clear and the paper is easy to follow.
- The proposed method outperforms all other TTA methods on four data domain adaptation settings.
- The experiments and ablation analysis are extensive.

**Weaknesses:**

- The paper lack of reports of runtime related results (time/memory consumption) to demonstrate the cost of introducing NPATT.
- The performance gains on real2real setting are not significant.
- It lacks insightful analysis for some results, e.g., why most TTA methods drop greatly on 3DFRONT→ScanNet setting compared with source only, while not on 3DFRONT→S3DIS setting.?
- The paper lacks of category-wise IoU results which could better demonstrates how the proposed method improve the overall performance compared with other methods.

**Questions:**

- How is the visualization made with features in Figure3? It’s better to also mention what x-axis and y-axis means.

---

> ### Author Response · Authors · 2024-11-20
>
> **Q1:** The paper lack of reports of runtime related results (time/memory consumption) to demonstrate the cost of introducing NGTTA.
>
> **R1:** Thank you for your suggestion. Regarding memory consumption and time, we reported the GFLOPs and TP (throughput (ins./sec.)) of different models in **Table 3 of the paper**. Please refer to the paper for more details.
> And from the results, we can see that the non-parametric geometric model we adjusted only increases memory consumption by approximately 0.9 GFLOPs.
>
> **Q2:** The performance gains on real2real setting are not significant.
>
> **R2:** Thank you for your question. Although the performance improvements brought by our method are not particularly outstanding compared to Sim2Real, **we believe that the main comparison should be made with previous TTA and UDA methods**. As shown, our approach also surpasses earlier TTA and UDA methods on real2real datasets, demonstrating the effectiveness of NGTTA. We will focus on improving the performance on real2real tasks in our future work.
>
> **Q3:** It lacks insightful analysis for some results, e.g., why most TTA methods drop greatly on 3DFRONT$\rightarrow$ScanNet setting compared with source only, while not on 3DFRONT$\rightarrow$S3DIS setting ?
>
> **R3:** Sorry for not explaining this phenomenon in the paper. We believe it is due to the characteristics of the dataset. First, previous TTA methods typically minimize the entropy loss in a straightforward manner, making them very sensitive to class imbalance and sample difficulty. In the case of ScanNet, the input scale is significantly larger than that of S3DIS, which exacerbates the class imbalance issue. Furthermore, the point cloud density and resolution in ScanNet are lower than those in S3DIS, making it inherently more challenging. Specifically, the average entropy for ScanNet is **0.62**, while for S3DIS, it is **0.31**, indicating that the model is less confident when dealing with ScanNet. These factors contribute to the difficulties in optimizing previous TTA methods on ScanNet, leading to a substantial decrease in performance.
>
> **Q4:** The paper lacks of category-wise IoU results which could better demonstrates how the proposed method improve the overall performance compared with other methods.
>
> **R4:**  Thank you for your suggestion. We have added the category IoU performance for 3DFRONT$\rightarrow$ScanNet, as shown in the following **Table R4**.  We can see that compared to TENT and Baseline, **a significant improvement of NGTTA is observed in the performance of minority and difficult classes**. For example, categories such as door, window, bookshelf, and desk have a quantity ratio of only 2\% to 4\%, and their original performance was poor. As discussed in the third point, previous TTA methods are quite sensitive to class imbalance and the difficulty level of samples. Consequently, after TENT optimization, training may collapse, resulting in an IoU of 0 for these categories. In contrast, NGTTA employs a Category-Balance Sampler to address the class-imbalance problem and distills the underlying manifold features of difficult samples using a non-parametric geometric model, leading to significantly improved performance in these categories.  **In the uploaded revised version of our paper, we have included this additional result in Appendix A.4.2 .**
>
> **Table R4: Category-Wise IoU Result on 3DFRONT$\rightarrow$ScanNet.**
> | **Method** | **wall** | **floor** | **cabinet** | **bed** | **chair** | **sofa** | **table** | **door** | **window** | **bookshelf** | **desk** | **mIoU** |
> |------------|----------|-----------|-------------|---------|-----------|----------|-----------|----------|------------|----------------|----------|----------|
> | Baseline   | 60.80    | 83.22     | 13.15       | 47.93   | 56.35     | 47.38    | 39.61     | 1.85     | 3.28       | 18.95          | 21.07    | 34.80    |
> | TENT       | 47.12    | 60.55     | 0.03        | 12.79   | 4.35      | 2.29     | 30.22     | 0        | 0          | 0              | 0        | 15.63    |
> | **NGTTA**  | 60.12    | 86.35     | 9.29        | 42.75   | 57.97     | 44.87    | 44.85     | 3.20     | 7.13       | 31.26          | 27.04    | **38.42** |
>
> **Q5:** How is the visualization made with features in Figure3? It’s better to also mention what x-axis and y-axis means.
>
> **R5:** The meaning of this figure is the visualization of sample feature distribution in the source domain and target domain, and each curve represents the distribution of **each feature channel**. So the X-axis represents the values of the feature distribution, while the Y-axis represents the probability densities of different values of the feature distribution.

---

> ### Author Response · Authors · 2024-11-25
> **End of Discussion Approaching**
>
> Dear Reviewer HLQ7:
>
> We sincerely thank you for taking the time to provide detailed comments. We have made corresponding revisions based on your questions and added a substantial amount of experiments.
>
> Given that the end of the discussion period is approaching, we would like to ask if there are any unclear aspects regarding our work. We are more than happy to provide further explanations as needed.
>
>
> Our work primarily explores Test Time Adaptation in point cloud scenes, conducted under the challenging SIm2Real setting, which is still an underexplored area. Our approach involves innovative attempts and achieves state-of-the-art performance. We hope that this work will be recognized and valued.
>
> We would like to express our gratitude once again for your feedback and suggestions, which have greatly helped us improve our work.
>
> Best and sincere wishes,
>
> The authors

---

> > ### Comment · Reviewer_HLQ7 · 2024-11-27
> >
> > Thank you for the detailed explanations. Most of my concerns and questions have been addressed.
> > I acknowledge the novelty in introducing a non-parametric encoder and the performance advantages demonstrated in the point cloud segmentation task. However, I would like to maintain my rating as I find the overall novelty and contribution maybe insufficient for an ICLR paper. My main points are as follows:
> > - While Test-Time Adaptation (TTA) is a more general setting and the most of the selected baselines are tailored to images, the proposed method is only evaluated on the point cloud segmentation. It would strengthen the work to demonstrate the generalizability of the proposed design (distillation, soft voting, CB sampler) on image datasets.
> > - The sim-to-real point cloud setting may lack practicality, as the primary goal of simulation is to facilitate easy data acquisition. Additionally, the performance improvement in the real-to-real setup is not significant. Evaluations on more challenging data gaps (e.g., ScanNet to KITTI) would be more compelling.
> > - The core claimed contribution of this work is the non-parametric model. However, the ablation studies focus on distillation, soft voting, and the CB sampler, which makes the emphasis on the main claimed contribution unclear. Ablation comparisons with parametric methods would better demonstrate the contribution.
> >
> > Given these observations, I believe this work might be more suitable for a CV-focused venue.

---

> > > ### Author Response · Authors · 2024-11-27
> > > **Response to  Reviewer HLQ7 (Part 1/2)**
> > >
> > > Thank you for your response. I hope the following answers will address your concerns:
> > >
> > > **1.For the First Point:**
> > >
> > > First of all, Test Time Adaptation (TTA) has primarily been applied in the image domain. However, with the advancement of point cloud analysis, an increasing number of researchers are focusing on applications in 3D data, such as [1-4]. These methods mainly explore object-level and outdoor object detection tasks, which do not align with indoor point cloud segmentation. **To our knowledge, we are the first to investigate the application of TTA in indoor point cloud segmentation**. Furthermore, since the development of TTA in the 3D domain is still in its infancy, **previously mentioned methods have largely compared with 2D TTA approaches.**  Therefore, our experimental setup is reasonable.  I hope this explanation can addresses your first concern.
> > >
> > > [1] Dastmalchi, Hamidreza, et al. "Test-Time Adaptation of 3D Point Clouds via Denoising Diffusion Models." arXiv preprint arXiv:2411.14495 (2024).
> > >
> > > [2]Shim, Hajin, Changhun Kim, and Eunho Yang. "CloudFixer: Test-Time Adaptation for 3D Point Clouds via Diffusion-Guided Geometric Transformation." European Conference on Computer Vision. Springer, Cham, 2025.
> > >
> > > [3]Chen, Zhuoxiao, et al. "DPO: Dual-perturbation optimization for test-time adaptation in 3d object detection." Proceedings of the 32nd ACM International Conference on Multimedia. 2024.
> > >
> > > [4]Lin, Hongbin, et al. "Fully Test-Time Adaptation for Monocular 3D Object Detection." arXiv preprint arXiv:2405.19682 (2024).
> > >
> > > **2.For the Second Point:**
> > >
> > > (a) **Sim2Real Setting**.  On the contrary, we believe that studying Sim2Real is highly necessary. **The ease of obtaining synthetic data has led to its increasing application in the 3D field for training purposes**. For instance, in autonomous driving, many companies, such as Tesla, utilize methods like world models to generate synthetic datasets. Additionally, the emerging field of embodied intelligence trains its models using synthetic data from simulators before transferring them to the real world. **Due to the differences between synthetic and real data, this has spurred the development of numerous methods [5,6]  focused on Sim2Real**, primarily in outdoor detection applications related to autonomous driving. To our knowledge, we are the first to investigate indoor Sim2Real TTA methods, which could be applied in areas such as embodied intelligence. Therefore, researching Sim2Real is indeed essential
> > >
> > >
> > > [5] Yuan, Zhimin, et al. "Density-guided Translator Boosts Synthetic-to-Real Unsupervised Domain Adaptive Segmentation of 3D Point Clouds." Proceedings of the IEEE/CVF Conference on Computer Vision and Pattern Recognition. 2024.
> > >
> > > [6]Yao, Yichen, et al. "HUNTER: Unsupervised Human-centric 3D Detection via Transferring Knowledge from Synthetic Instances to Real Scenes." Proceedings of the IEEE/CVF Conference on Computer Vision and Pattern Recognition. 2024.
> > >
> > > (b) **Real2Real Setting.**   We believe that the performance improvement of Real2Real is sufficient. First, we can enhance the Baseline by 3-4 \%mIoU. Furthermore, compared to the state-of-the-art 2D TTA method Adacontrast, we achieve improvements of 0.5\% and 1.1 \%mIoU while utilizing less computational resources and achieving faster training speeds. Additionally, NGTTA can also match the performance of UDA methods. Therefore, we consider the performance improvement to be adequate.
> > >
> > > (c) **ScanNet2KITTI** .  As we stated in the Rebuttal,it is impossible to fully evaluate the performance of Test Time Adaptation methods for the transition from indoor to outdoor due to the lack of shared label space. **Secondly, we conducted simple experiments from 3D FRONT to SemanticKITTI, where the domain gap is larger than that from ScanNet to KITTI**. This is because the differences between synthetic and real data are more pronounced; for example, the performance of 3D FRONT to ScanNet is significantly lower than that of S3DIS to ScanNet. **Therefore, the experiments we provided from 3D FRONT to SemanticKITTI are more convincing, and the performance improvements indicate the great potential of NG TTA for the task of transferring from indoor to outdoor environments.**

---

> > > > ### Author Response · Authors · 2024-11-27
> > > > **Response to  Reviewer HLQ7 (Part 2/2)**
> > > >
> > > > **3.For the Third Point :**
> > > >
> > > >
> > > > On the contrary, most of the ablation experiments we conducted demonstrate the importance of the non-parametric geometric model.
> > > >
> > > > (1) The first ablation experiment is presented in Table 4 of the main text, **which shows that the non-parametric geometric model outperforms existing pre-trained parametric models**, evidencing its strong generalization capability.
> > > >
> > > > (2) The second ablation experiment, detailed in Table 5, investigates the effects of distillation, soft voting, and sampling mechanisms. In this case,**the distillation is entirely dominated by the non-parametric geometric model**, which improves the mIoU by 3\%.  **The soft voting, which involves both model features and non-parametric geometric features**, leads to a 2\% improvement in mIoU. This sufficiently demonstrates that the non-parametric geometric model is a key factor in enhancing the performance of NGTTA.
> > > >
> > > > (3) The third ablation experiment is presented in Table 6 of the main text, which shows that in soft voting, **combining non-parametric geometric features with model features leads to a 0.5\% improvement in mIoU compared to using model features alone**, thereby demonstrating the importance of the non-parametric geometric model.
> > > >
> > > > I hope the reviewers can revisit the paper and assess the significance and innovation of the non-parametric geometric model.
> > > >
> > > > **4:**
> > > >
> > > > We appreciate the reviewers' feedback and suggestions, which have helped us improve our approach. To the best of our knowledge, NG TTA is the first method to apply Test Time Adaptation (TTA) to indoor segmentation tasks and achieve state-of-the-art performance. With the clarifications provided above, we hope the reviewers can reevaluate the contributions of our method.

---

> > > > > ### Author Response · Authors · 2024-11-30
> > > > > **Did our answers solve all your questions?**
> > > > >
> > > > > Dear Reviewer HLQ7:
> > > > >
> > > > > Please allow us to sincerely thank you again for your constructive comments and valuable feedback. We believe our latest response has addressed your points, but please let us know if there is anything else we can clarify or assist with. We are more than happy to answer any further questions during the discussion period. Your feedbacks are truly valued!
> > > > >
> > > > > Best and sincere wishes,
> > > > >
> > > > > The authors

---

> > > > > > ### Author Response · Authors · 2024-12-01
> > > > > > **Request review of recent updates**
> > > > > >
> > > > > > Dear Reviewer HLQ7:
> > > > > >
> > > > > > As the discussion is coming to a close, we sincerely hope to receive your feedback on our recent clarifications.
> > > > > >
> > > > > >  In response to the three concerns you raised, we have thoroughly clarified the rationale behind the TTA and Sim2Real experimental setups, supported by extensive literature and developments in the current field. Additionally, we have explained that the ablation studies on the non-parametric geometric model actually constitute a significant portion of our paper.
> > > > > >
> > > > > > We greatly appreciate the time you have taken to read our responses and look forward to your feedback.
> > > > > >
> > > > > > Best and sincere wishes,
> > > > > >
> > > > > > The authors

---

### Meta-Review · Area_Chair_mzuT · 2024-12-19

**Metareview:**

This paper presents a test-time adaptation method, NGTTA, which is driven by non-parametric geometry to address the training collapse issue caused by domain gaps when applied to complex 3D scenes for point cloud segmentation. The main idea is to model the distribution of non-parametric geometry by PointNN for target data as an *intermediate domain*. A category-balance sampler has been adopted to address the class imbalance issue in semantic segmentation. Easy samples are used to compute the entropy minimization loss and pseudo-label prediction loss. Furthermore, soft voting is not only conducted among nearest neighbors in the model feature space but also within the geometric space to mitigate the accumulation of errors. In the early stages of training, the geometric features of hard samples are distilled into the source domain model to accelerate convergence to the intermediate domain.

Reviewers have raised concerns regarding the paper's technical contributions. They noted that several components are simply adapted from existing 2D TTA methods, which appear easily applicable to the 3D segmentation tasks, as shown by the experimental results. The contribution of employing the non-parametric model seems limited, as it is heavily based on PointNN. The area chair has also gone through the paper and shares similar concerns. Although the authors included extra experimental results in their revision, the concern about technical contributions remains unresolved. The paper's final ratings are 5, 5, 6, and 6 after the reviewer discussion phase, suggesting that the paper lacks adequate support for acceptance.

**Additional Comments On Reviewer Discussion:**

In addition to the concern about technical contributions, here are some further comments on the reviewer discussion:
> Other issues that still exist during post-rebuttal are the hyperparameters, their choice/bias, and ablations of alternative combinations (although substantial experiments have been added). The writing also remains an issue, as it still lacks clarity. The manuscript needs further experiments and discussions of results; however, improvements in the clarity of writing could enhance its value for the field.

The paper by Wang et al., titled "Backpropagation-free Network for 3D Test-time Adaptation," published in CVPR2024, is also relevant. Although it addresses a different application than indoor scene segmentation, it may be more closely related to the topic than other 2D test-time adaptation methods.

---

### Decision · Program_Chairs · 2025-01-22

Reject